# JADAI: Jointly Amortizing Adaptive Design and Bayesian Inference

Niels Bracher [* 1]   Lars Kühmichel [* 2]   Desi R. Ivanova [3]   Xavier Intes [1]   Paul-Christian Bürkner [2]
Stefan T. Radev [1]

## Abstract

We consider problems of parameter estimation where design variables can be actively optimized to maximize information gain. To this end, we introduce JADAI, a framework that jointly amortizes Bayesian adaptive design and inference by training a policy, a history network, and an inference network end-to-end. The networks minimize a generic loss that aggregates incremental reductions in posterior error along experimental sequences without density evaluations. Inference networks are instantiated with diffusion models that can approximate high-dimensional and multimodal posteriors at every experimental step. JADAI achieves superior or competitive performance across adaptive design benchmarks.

## 1. Introduction

Many scientific and engineering questions concern *unobserved* properties of real-world systems: cosmological parameters governing large-scale structure (Hahn et al., 2024), biophysical parameters in mechanistic neural models (Gonçalves et al., 2020), or epidemiological parameters driving disease dynamics (Radev et al., 2021). In many such settings, we can design simulators that, given hypothesized parameters $\boldsymbol{\theta}$, can generate synthetic observations $\mathbf{x}$. However, inverting these simulators to recover the parameters from observations is oftentimes a computational ordeal.

Simulation-based inference (SBI) addresses this inverse problem by approximating the posterior over parameters from simulated pairs $(\boldsymbol{\theta}, \mathbf{x})$, typically using a neural network (Cranmer et al., 2020; Zammit-Mangion et al., 2025; Deistler et al., 2025). Modern SBI employs generative models, such as flow-matching (Wildberger et al., 2023), diffu-

sion (Sharrock et al., 2024), or consistency models (Schmitt et al., 2024) to represent complex, multimodal posteriors in high-dimensional parameter spaces (Arruda et al., 2025). Once trained, these *amortized models* can be applied to real observations to estimate the unobserved parameters.

In many of these applications, however, we are not only handed data, but can also *actively control* how data is collected or how the system is perturbed. Experimental protocols, stimulus sequences, or public health interventions are often parameterized by design variables $\boldsymbol{\xi}$ that strongly influence how informative the resulting observations $\mathbf{x}$ are about $\boldsymbol{\theta}$. This experimental adaptability turns the inference problem into a two-fold question: (i) how to infer $\boldsymbol{\theta}$ from a given dataset, and (ii) how to choose designs $\boldsymbol{\xi}$ that make inferences as informative as possible.

Bayesian experimental design (BED) formalizes an answer to the second question by selecting designs that maximize an expected utility, most commonly the expected information gain (EIG) about $\boldsymbol{\theta}$ (Lindley, 1956). In Bayesian adaptive design (BAD), this becomes a sequential problem in which a *designer* (e.g., a neural network) proposes designs based on previous decisions and data acquisitions. This idea underlies recent works on deep Bayesian adaptive design, which learn global history-dependent policies directly from simulations (Foster et al., 2021; Ivanova et al., 2021; Blau et al., 2022; Huang et al., 2026), rather than solving a new optimization problem from scratch for every experiment.

Despite this progress, BAD and SBI have largely evolved in parallel. Policy-based BAD approaches typically focus on proposing good designs, delegating posterior inference to slow (non-amortized) methods or restricting it to relatively simple parametric families in low-dimensional settings. Conversely, SBI methods usually work with fixed designs and do not directly optimize how observations are acquired. As a result, design and inference are typically treated as separate tasks (see Section 4 for related work).

In this paper, we introduce JADAI, a framework for **J**ointly **A**daptive **D**esign and **A**mortized Bayesian **I**nference. Our framework makes the following contributions:

- It jointly amortizes adaptive design and posterior inference via a general utility that aggregates incremental

---

[*]Equal contribution  [1]Rensselaer Polytechnic Institute, USA [2]TU Dortmund University, Germany [3]University of Oxford, UK. Correspondence to: Niels Bracher <brachn@rpi.edu>, Lars Kühmichel <larskuedev@gmail.com>.

*Proceedings of the 43rd International Conference on Machine Learning*, Seoul, South Korea. PMLR 306, 2026. Copyright 2026 by the author(s).

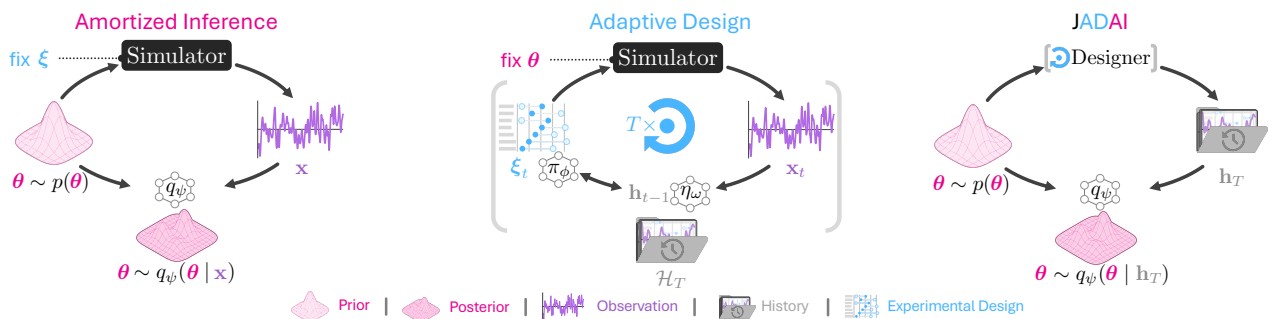

*Figure 1.* Overview of amortized SBI, BAD, and our proposed JADAI framework. *Left:* Amortized SBI, where a neural posterior estimator $q_\psi(\boldsymbol{\theta} \mid \mathbf{x})$ is trained on simulator pairs $(\mathbf{x}, \boldsymbol{\theta})$ under a fixed design $\boldsymbol{\xi}$. *Middle:* Amortized BAD, where a designer runs a full $T$-step rollout for a fixed parameter $\boldsymbol{\theta}$: a policy $\pi_\phi$ maps the history state $\mathbf{h}_{t-1} = \eta_\omega(\{(\boldsymbol{\xi}_k, \mathbf{x}_k)\}_{k=1}^{t-1})$ to a new design $\boldsymbol{\xi}_t$, and the simulator returns $\mathbf{x}_t$, iteratively forming the history $\mathcal{H}_T = \{(\boldsymbol{\xi}_t, \mathbf{x}_t)\}_{t=1}^T$. *Right:* JADAI embeds this designer (middle block) into the amortized SBI loop and jointly trains $\pi_\phi$, $\eta_\omega$, and $q_\psi$ end-to-end, amortizing both experimental design and posterior inference across rollouts and experiments.

improvements end-to-end without density evaluations.

- It incorporates modern diffusion-based generative models, providing amortized high-dimensional and multi-modal posteriors at every experimental step.

- It achieves superior or competitive performance across a range of adaptive design benchmarks.

## 2. Background

### 2.1. Simulators as statistical models

For the purpose of our discussion, a simulator generates observables $\mathbf{x} \in \mathcal{X}$ as a function of unknown parameters $\boldsymbol{\theta} \in \Theta$, design variables $\boldsymbol{\xi} \in \Xi$ and random program states $\mathbf{z} \in \mathcal{Z}$:

$$\mathbf{x} = \mathrm{Sim}(\boldsymbol{\theta}, \mathbf{z}; \boldsymbol{\xi}) \quad \text{with} \quad \boldsymbol{\theta} \sim p(\boldsymbol{\theta}), \ \mathbf{z} \sim \mathrm{RNG}(\cdot) \quad (1)$$

The *forward problem* in (1) is typically well-understood through a mathematical model. The *inverse problem*, however, is typically much harder, and forms the crux of Bayesian inference: for a given design $\boldsymbol{\xi}$, estimate the unknowns $\boldsymbol{\theta}$ from observables $\mathbf{x}$ via the *posterior distribution*:

$$p(\boldsymbol{\theta} \mid \mathbf{x}; \boldsymbol{\xi}) \propto p(\boldsymbol{\theta}) \, p(\mathbf{x} \mid \boldsymbol{\theta}; \boldsymbol{\xi}). \quad (2)$$

However, when working with complex simulators, the *likelihood* $p(\mathbf{x} \mid \boldsymbol{\theta}, \boldsymbol{\xi})$ needed for proper statistical inference cannot be directly evaluated. In these cases, a posterior estimator $q(\boldsymbol{\theta} \mid \mathbf{x}; \boldsymbol{\xi})$ needs to be constructed from simulations of (1) alone. This is the central idea of simulation-based inference (SBI; Cranmer et al., 2020; Diggle & Gratton, 1984).

### 2.2. Neural simulation-based inference

In neural SBI, synthetic pairs $(\boldsymbol{\theta}_n, \mathbf{x}_n)$ obtained via (1) are used to train a (generative) neural network. The network can approximate either the intractable posterior, likelihood, or both. The network is then applied to unlabeled real observations $\mathbf{x}^o$, essentially solving a sim2real problem. Neural SBI is compatible with any off-the-shelf generative model, such as normalizing flows (Rezende & Mohamed, 2015), flow matching (Lipman et al., 2023), or diffusion models (Song & Ermon, 2019). In this work, we focus specifically on *amortized methods*, which train a single (global) posterior estimator $q(\boldsymbol{\theta} \mid \mathbf{x}; \boldsymbol{\xi})$ that remains valid for any $\mathbf{x}^o \sim p^*(\mathbf{x}^o)$ (see Figure 1, *left*).

### 2.3. Bayesian experimental design

SBI methods are typically developed and applied for a fixed design $\boldsymbol{\xi}$. Ideally, we want to choose the design that maximizes the amount of information we expect to gain from our subsequent observations. This is the goal of Bayesian experimental design (BED; Lindley, 1956; Chaloner & Verdinelli, 1995; Rainforth et al., 2024).

**Realized information gain.** Suppose we run an experiment with design $\boldsymbol{\xi}$ and obtain an actual observation $\mathbf{x}^o$. The *realized* information gain (IG) is defined as the reduction in entropy from the prior $p(\boldsymbol{\theta})$ to the posterior $p(\boldsymbol{\theta} \mid \mathbf{x}^o; \boldsymbol{\xi})$:

$$\mathrm{IG}(\mathbf{x}^o, \boldsymbol{\xi}) := \mathbb{H}\left[p(\boldsymbol{\theta})\right] - \mathbb{H}\left[p(\boldsymbol{\theta} \mid \mathbf{x}^o; \boldsymbol{\xi})\right], \quad (3)$$

where $\mathbb{H}[p]$ denotes the Shannon entropy of $p$. Crucially, the IG can be evaluated only *after* observing the outcome $\mathbf{x}^o$ and fitting a posterior $p(\boldsymbol{\theta} \mid \mathbf{x}^o; \boldsymbol{\xi})$.

**Expected information gain.** Before running the experiment, however, the outcome $\mathbf{x}$ is unknown. Thus, the realized IG (3) cannot be used to select a design $\boldsymbol{\xi}$. Instead, BED aims to select the design that maximizes the *expected*

information gain (EIG):

$$\text{EIG}(\boldsymbol{\xi}) := \mathbb{E}_{p(\mathbf{x}|\boldsymbol{\xi})}\left[\text{IG}(\mathbf{x}, \boldsymbol{\xi})\right] \tag{4}$$

$$= \mathbb{E}_{p(\boldsymbol{\theta})p(\mathbf{x}|\boldsymbol{\theta},\boldsymbol{\xi})}\left[\log p(\boldsymbol{\theta} \mid \mathbf{x}; \boldsymbol{\xi}) - \log p(\boldsymbol{\theta})\right], \tag{5}$$

where $p(\mathbf{x} \mid \boldsymbol{\xi}) = \int p(\mathbf{x} \mid \boldsymbol{\theta}, \boldsymbol{\xi})\, p(\boldsymbol{\theta})\, \mathrm{d}\boldsymbol{\theta}$ denotes the prior predictive distribution. The second expression shows that $\text{EIG}(\boldsymbol{\xi})$ is the mutual information between $\boldsymbol{\theta}$ and $\mathbf{x}$ under design $\boldsymbol{\xi}$. Maximizing the EIG, therefore, selects designs that, *on average*, are expected to yield the most informative observations about $\boldsymbol{\theta}$.

## 2.4. Bayesian adaptive design

BED is particularly attractive when experiments entail a sequence of design decisions $\boldsymbol{\xi}_1, \dots, \boldsymbol{\xi}_T$, where each decision can make use of previous observations (Rainforth et al., 2024). Let $\mathcal{H}_{t-1} = \{(\boldsymbol{\xi}_k, \mathbf{x}_k)\}_{k=1}^{t-1}$ denote the raw experimental history up to step $t$. At each such step, we can update our beliefs $p(\boldsymbol{\theta} \mid \mathcal{H}_{t-1})$ based on all information gathered so far, and then choose the *next design* $\boldsymbol{\xi}_t$ by maximizing the *incremental* expected information gain:

$$\text{EIG}(\boldsymbol{\xi}_t \mid \mathcal{H}_{t-1}) :=$$
$$\mathbb{E}_{p(\boldsymbol{\theta}|\mathcal{H}_{t-1})\, p(\mathbf{x}_t|\boldsymbol{\theta},\boldsymbol{\xi}_t)}\left[\log \frac{p(\boldsymbol{\theta} \mid \mathcal{H}_{t-1}, \mathbf{x}_t; \boldsymbol{\xi}_t)}{p(\boldsymbol{\theta} \mid \mathcal{H}_{t-1})}\right]. \tag{6}$$

This expression is simply the standard EIG (5), but evaluated using the *current* posterior as the prior for the next step. Thus, BAD selects designs by maximizing (6) at each step $t$, with $\mathcal{H}_0 = \varnothing$ (see Figure 1, *middle*).

Foster et al. (2021) introduced a *policy network* $\pi_\phi$ that predicts the next design from history, $\pi_\phi(\mathcal{H}_{t-1}) = \boldsymbol{\xi}_t$, and showed that the decision process can be amortized over all $T$ steps by maximizing the *total* EIG (TEIG):

$$\text{TEIG}(\pi_\phi) := \mathbb{E}\left[\sum_{t=1}^{T} \text{EIG}\left(\pi_\phi(\mathcal{H}_{t-1}) \mid \mathcal{H}_{t-1}\right)\right] \tag{7}$$

$$= \mathbb{E}\left[\log p(\mathcal{H}_T \mid \boldsymbol{\theta}, \pi_\phi) - \log p(\mathcal{H}_T \mid \pi_\phi)\right] \tag{8}$$

$$= \mathbb{E}\left[\log p(\boldsymbol{\theta} \mid \mathcal{H}_T, \pi_\phi) - \log p(\boldsymbol{\theta})\right], \tag{9}$$

where all expectations are under $p(\boldsymbol{\theta})\, p(\mathcal{H}_T \mid \boldsymbol{\theta}, \pi_\phi)$. In the next section, we extend this framework to the typical SBI setting, and jointly amortize both the design policy and posterior inference across experimental time.

# 3. Method

## 3.1. Setup

Our framework builds on the SBI and BAD setting from Section 2 (see also Figure 1, *right*). To represent the history in a fixed-dimensional form, we introduce a summary network

$\eta_\omega$ that maps the history sequence $\mathcal{H}_t = \{(\boldsymbol{\xi}_k, \mathbf{x}_k)\}_{k=1}^{t}$ to a summary statistic

$$\mathbf{h}_t = \eta_\omega(\mathcal{H}_t), \qquad t = 0, \dots, T, \tag{10}$$

with $\mathbf{h}_0 = \mathbf{0}$ corresponding to the empty history state. As in amortized policy-based BAD (Foster et al., 2021; Ivanova et al., 2021), a deterministic policy network $\pi_\phi$ selects the next design based on the current summary, $\boldsymbol{\xi}_t = \pi_\phi(\mathbf{h}_{t-1})$ at each step $t \geq 1$.

Inference is performed by a neural posterior estimator $q_\psi(\boldsymbol{\theta} \mid \mathbf{h}_t)$, which takes the summary $\mathbf{h}_t$ as input and approximates the intractable posterior $p(\boldsymbol{\theta} \mid \mathbf{h}_t)$. For a simulated pair $(\boldsymbol{\theta}, \mathbf{h}_t)$ we define the per-step training loss

$$\ell_t(\boldsymbol{\theta}, \mathbf{h}_t; \phi, \omega, \psi) := \mathcal{L}_{\text{post}}(\boldsymbol{\theta}, \mathbf{h}_t; \phi, \omega, \psi), \tag{11}$$

where $\mathcal{L}_{\text{post}}$ is the posterior loss induced by the posterior estimator (e.g., score matching or flow matching; see Section A.2 and Section A.3 for details). Although $q_\psi$ is the only density estimator, $\ell_t$ depends on all parameters $(\phi, \omega, \psi)$ through the summary $\mathbf{h}_t = \eta_\omega(\{(\boldsymbol{\xi}_k, \mathbf{x}_k)\}_{k=1}^{t})$ and the designs $\boldsymbol{\xi}_k = \pi_\phi(\mathbf{h}_{k-1})$. We treat $\ell_t$ as a *shared training signal* to update the policy $\pi_\phi$, the summary network $\eta_\omega$, and the posterior estimator $q_\psi$. For notational convenience, we will usually suppress the explicit dependence on $(\phi, \omega, \psi)$ and write $\ell_t$ instead of $\ell_t(\boldsymbol{\theta}, \mathbf{h}_t; \phi, \omega, \psi)$.

## 3.2. Optimization objective

To obtain a tractable training objective, we use the Barber-Agakov variational lower bound (Barber & Agakov, 2003; Foster et al., 2019; Blau et al., 2023) and replace the true posterior $p(\boldsymbol{\theta} \mid \mathbf{h}_T)$ by a neural estimator $q_\psi(\boldsymbol{\theta} \mid \mathbf{h}_T)$ in (9):

$$\text{TEIG}(\pi_\phi) \geq \mathbb{E}_{p(\boldsymbol{\theta}, \mathbf{h}_T | \pi_\phi, \eta_\omega)}\left[\log \frac{q_\psi(\boldsymbol{\theta} \mid \mathbf{h}_T)}{p(\boldsymbol{\theta})}\right], \tag{12}$$

with equality if and only if $q_\psi(\boldsymbol{\theta} \mid \mathbf{h}_T) = p(\boldsymbol{\theta} \mid \mathbf{h}_T)$ almost everywhere.

For intuition, we first consider the case when $q_\psi$ is a normalized density model, i.e., a normalizing flow with $\ell_t^{\text{flow}}(\boldsymbol{\theta}, \mathbf{h}_t) := -\log q_\psi(\boldsymbol{\theta} \mid \mathbf{h}_t)$ as in Blau et al. (2023). Additionally, define the per-step loss $\ell_{-1}^{\text{flow}}(\boldsymbol{\theta}) := -\log p(\boldsymbol{\theta})$. Then the logarithm in (12) admits the decomposition

$$\log \frac{q_\psi(\boldsymbol{\theta} \mid \mathbf{h}_T)}{p(\boldsymbol{\theta})} = \sum_{t=1}^{T} \log \frac{q_\psi(\boldsymbol{\theta} \mid \mathbf{h}_t)}{q_\psi(\boldsymbol{\theta} \mid \mathbf{h}_{t-1})} + \log \frac{q_\psi(\boldsymbol{\theta} \mid \mathbf{h}_0)}{p(\boldsymbol{\theta})} \tag{13}$$

$$= \sum_{t=0}^{T} \ell_{t-1}^{\text{flow}} - \ell_t^{\text{flow}}. \tag{14}$$

Hence, for a normalized density model, the variational TEIG bound can be written as a telescoping sum of per-step differences in information content. However, the use of normalizing flows can be restrictive in practice (Chen et al., 2025),

and we want our framework to generalize beyond inference models with exact log-density computation.

Thus, we suggest using the generic quantity $\ell_{t-1} - \ell_t$ as a proxy for the incremental information gained from step $t - 1$ to $t$, even when $q_\psi$ is approximated with an implicit model for which the normalized log-posterior is not directly available. For example, a diffusion model formulation of (14) entails the difference in posterior scores as a summand:

$$\nabla_{\boldsymbol{\theta}_\tau} \log p(\boldsymbol{\theta}_\tau \mid \mathbf{h}_{t-1}) - \nabla_{\boldsymbol{\theta}_\tau} \log p(\boldsymbol{\theta}_\tau \mid \mathbf{h}_t), \quad (15)$$

where $\tau$ denotes diffusion time (see Section A.2). In that case, the Barber-Agakov interpretation (Barber & Agakov, 2003) is no longer the same, but the resulting objective still encourages trajectories along which the posterior loss decreases, as demonstrated by our experiments in Section 5. For a more detailed discussion of the relation to the Barber–Agakov bound, see Appendix Section A.1.

This motivates the definition of the general scalar utility

$$u_T(\boldsymbol{\theta}, \mathbf{h}_{0:T}) := \sum_{t=0}^{T} \left( \ell_{t-1}^* - \ell_t \right), \qquad \ell_{-1} := 0, \quad (16)$$

where $\ell_t^* = \mathtt{detach}(\ell_t)$ denotes the same loss value with gradients stopped. Our final training objective minimizes the negative expected utility

$$\mathcal{L}(\phi, \psi, \omega) := \mathbb{E}_{p(\boldsymbol{\theta})p(\mathbf{h}_{0:T} \mid \boldsymbol{\theta}, \pi_\phi, \eta_\omega)} \left[ -u_T(\boldsymbol{\theta}, \mathbf{h}_{0:T}) \right], \quad (17)$$

using mini-batch gradient descent on $\mathcal{L}$. Since the sum in (16) reduces to $u_T(\boldsymbol{\theta}, \mathbf{h}_{0:T}) = -\ell_T(\boldsymbol{\theta}, \mathbf{h}_T)$, maximizing $u_T$ is equivalent to minimizing the final posterior loss. However, because the baseline terms $\ell_{t-1}^*$ are treated as constants during backpropagation, *telescoping does not apply to gradients*. For any parameter block, say $\phi$, we obtain

$$\frac{\partial \mathcal{L}}{\partial \phi} \approx -\frac{\partial u_T}{\partial \phi} = \sum_{t=0}^{T} \frac{\partial \ell_t}{\partial \phi}. \quad (18)$$

Thus, the gradient direction aggregates contributions from *all* intermediate losses $\ell_t(\boldsymbol{\theta}, \mathbf{h}_t)$ and pushes the networks $(\pi_\phi, \eta_\omega, q_\psi)$ towards improving the posterior approximation at every step along the experimental history sequence for $t \geq 1$ and an initial approximation to the prior at $t = 0$.

In practice, the expectation in (17) is approximated via Monte Carlo. For each training instance in the mini-batch, we sample a parameter $\boldsymbol{\theta} \sim p(\boldsymbol{\theta})$, initialize the empty history state $\mathbf{h}_0$, and evaluate the initial loss $\ell_0(\boldsymbol{\theta}, \mathbf{h}_0)$ so that $q_\psi$ learns to approximate the prior at $t = 0$. We then iteratively generate the *history rollout* to eventually obtain $-u_T(\boldsymbol{\theta})$ and update $(\phi, \omega, \psi)$ using its averaged gradient. The full training procedure is outlined in Algorithm 1.

---

**Algorithm 1** Joint amortization of policy and posterior networks

**Input:** prior $p(\boldsymbol{\theta})$; simulator $\mathrm{Sim}(\boldsymbol{\theta}, \boldsymbol{\xi})$; networks: summary $\eta_\omega$, policy $\pi_\phi$, posterior $q_\psi$; max horizon $T$; schedules $R(n; T)$, $\rho(n)$; window $W$
**Output:** trained parameters $(\phi, \omega, \psi)$

1 **for** training iteration $n = 1, 2, \dots$ **do**
2    $\boldsymbol{\theta} \sim p(\boldsymbol{\theta})$        *(sample ground truth)*
3    $r \sim \mathcal{U}\{1, \dots, R(n; T)\}$    *(sample rollout length)*
4    $\mathcal{H}_0 \leftarrow \varnothing$          *(initialize history)*
5    $\mathbf{h}_0 \leftarrow \mathbf{0}$         *(initialize summary)*
6    $\ell_0 \leftarrow \mathcal{L}_{\mathrm{post}}(\boldsymbol{\theta}, \mathbf{h}_0)$       *(prior loss)*
7    $u_0 \leftarrow -\ell_0$         *(update utility)*
8    $\ell_0^* \leftarrow \mathtt{detach}(\ell_0)$
9    **for** $t = 1$ **to** $r$ **do**    *(rollout & loss aggregation)*
10      With prob. $\rho_n$ set $\boldsymbol{\xi}_t \sim p(\boldsymbol{\xi})$, otherwise
       $\boldsymbol{\xi}_t \leftarrow \pi_\phi(\mathtt{detach}(\mathbf{h}_{t-1}))$  *(prior mix-in or policy)*
11      $\mathbf{x}_t \sim \mathrm{Sim}(\boldsymbol{\theta}, \boldsymbol{\xi}_t)$   *(simulate measurement)*
12      $\mathcal{H}_t \leftarrow \mathcal{H}_{t-1} \cup (\boldsymbol{\xi}_t, \mathbf{x}_t)$   *(update history)*
13      $\mathbf{h}_t \leftarrow \eta_\omega(\mathcal{H}_t)$        *(summarize)*
14      $\ell_t \leftarrow \mathcal{L}_{\mathrm{post}}(\boldsymbol{\theta}, \mathbf{h}_t)$    *(posterior loss)*
15      $u_t \leftarrow u_{t-1} + (\ell_{t-1}^* - \ell_t)$   *(update utility)*
16      $\ell_t^* \leftarrow \mathtt{detach}(\ell_t)$
17      **if** $t > W$ **then**   *(optional: truncated BPTT)*
18        $\mathtt{detach} \{(\boldsymbol{\xi}_k, \mathbf{x}_k)\}_{k=1}^{t-W} \in \mathcal{H}_t$
19    $\mathcal{L} \leftarrow -u_r$
20    update $(\phi, \omega, \psi)$ using $\nabla \mathcal{L}$    *(gradient step)*

---

### 3.3. Amortized design and inference

At test time, we freeze $(\phi, \omega, \psi)$ and use the learned triple $(\pi_\phi, \eta_\omega, q_\psi)$ by mirroring the rollout described above, but without loss evaluation or gradient updates. We initialize the empty history sequence and summary state $\mathbf{h}_0 = \mathbf{0}$ and, for $t = 1, \dots, T$, repeatedly predict the next design $\boldsymbol{\xi}_t = \pi_\phi \left( \mathbf{h}_{t-1} = \eta_\omega(\{(\mathbf{x}_k^o, \boldsymbol{\xi}_k)\}_{k=1}^{t-1}) \right)$ and make a new observation $\mathbf{x}_t^o \sim \mathrm{Experiment}(\boldsymbol{\xi}_t)$ by running the experiment. At any step, we could query the approximate posterior $q_\psi(\boldsymbol{\theta} \mid \mathbf{h}_t)$; in the diffusion case, this corresponds to conditional sampling described in Section A.2 with $\mathbf{h}_t$ as conditioning input. After the final step $T$, we obtain our best posterior approximation $q_\psi(\boldsymbol{\theta} \mid \mathbf{h}_T)$ given all measurements collected under the learned design policy.

### 3.4. Training in practice

**Avoiding nested backpropagation through time.** In the naive implementation of (17), each design is generated from the previous summary, $\boldsymbol{\xi}_t = \pi_\phi(\mathbf{h}_{t-1})$, and each summary is computed from all past tokens, $\mathbf{h}_t = \eta_\omega(\{(\boldsymbol{\xi}_k, \mathbf{x}_k)\}_{k=1}^t)$. As a result, the forward graph for $\boldsymbol{\xi}_t$ contains an increasingly long chain over time: $\boldsymbol{\xi}_1$ depends only on $\mathbf{h}_0$, $\boldsymbol{\xi}_2$ depends on $(\mathbf{h}_0, \boldsymbol{\xi}_1, \mathbf{x}_1, \mathbf{h}_1)$, $\boldsymbol{\xi}_3$ depends on $(\mathbf{h}_0, \boldsymbol{\xi}_1, \mathbf{x}_1, \mathbf{h}_1, \boldsymbol{\xi}_2, \mathbf{x}_2, \mathbf{h}_2)$, and so on. When backpropagating from a final loss $\ell_T(\boldsymbol{\theta}, \mathbf{h}_T)$, gradients therefore have to traverse both the rollout over time and, for each token $(\boldsymbol{\xi}_t, \mathbf{x}_t)$, an internally time-unrolled subgraph through earlier summaries and policy evaluations. In effect, this yields a nested form of back-

propagation through time (BPTT) whose depth grows on the order of $1 + 2 + \cdots + T$, and tightly couples the history representation to the policy via a cyclic gradient path (history state $\rightarrow$ policy $\rightarrow$ history state).

To avoid this feedback loop, we generate designs from a detached history state, $\boldsymbol{\xi}_t = \pi_\phi(\mathbf{h}_{t-1}^*)$. Forward passes are unchanged, but gradients from the posterior losses can no longer flow back into the history through the policy inputs. In other words, the summary network $\eta_\omega$ is trained only via its influence on the posterior losses $\ell_t(\boldsymbol{\theta}, \mathbf{h}_t)$, while the policy $\pi_\phi$ is trained via the effect of its designs on future losses, through the tokens $(\boldsymbol{\xi}_t, \mathbf{x}_t)$ that enter the summaries. Importantly, the history state itself is not detached from the loss that evaluates it: each $\ell_t(\boldsymbol{\theta}, \mathbf{h}_t)$ still backpropagates into $\eta_\omega$ through $\mathbf{h}_t$, so the history network receives direct feedback on the quality of its stepwise encoding.

A second, distinct stop-gradient appears in the utility (16): when a posterior loss is reused as the baseline in the next summand, we use $\ell_t^* = \text{detach}(\ell_t)$. This prevents the telescoping scalar objective from also telescoping in the gradient, so the update retains posterior-training signal from every rollout step (see Section A.1). Conceptually, this enforces the role of $\mathbf{h}_t$ as a *learned summary statistic* that is optimized to be maximally informative about the design-aware posterior $p(\boldsymbol{\theta} \mid \mathbf{h}_t)$ (Radev et al., 2020; Chen et al., 2023). The policy processes these approximately sufficient summaries to propose useful designs.

The above modifications remove the nested BPTT structure and leaves a single, simpler gradient path through the history, computationally as memory-intensive as standard BPTT in recurrent models. On top of this, we can optionally limit gradient propagation through time by detaching tokens older than a fixed window size $W$, so that losses at step $t$ only backpropagate through the most recent $W$ $(\boldsymbol{\xi}, \mathbf{x})$-pairs. This is analogous to truncated BPTT in RNNs and provides a simple mechanism to keep memory and compute manageable for longer rollouts or larger architectures (see Section 5.1 and Appendix B).

**Scheduling and sampling the rollout length.** Because the policy, summary, and posterior networks are mutually dependent, very long rollouts early in training can hinder learning: even the $t = 0$ posterior $q_\psi(\boldsymbol{\theta} \mid \mathbf{h}_0)$ must first learn to match the prior $p(\boldsymbol{\theta})$ before later decisions become meaningful. We therefore use a steep curriculum on the rollout length ($< 1\%$ of the total training time) via a monotone schedule on the current maximum rollout length $R(n) \leq T$ at training iteration $n$ that gradually increases towards $T$, and then sample the actual rollout length as $r \sim \mathcal{U}(1, R(n))$. For that iteration, the corresponding utility $u_r$ from Eq. (16) gets truncated at $t = r$. Sampling the $r$ reweights training toward shorter rollout regimes, where

fewer observations are available and posterior inference is typically harder. This is useful when the learned model should provide informative posteriors at intermediate stopping times. If only terminal performance at a known horizon $T$ is important, training with a fixed rollout length can be preferable (see Appendix B).

**Exploration via design prior mix-in.** Complementary to the rollout-length curriculum, we also use a curriculum on how designs are chosen during training. Early in training, the policy network tends to propose designs in a narrow region of the design space, so relying on $\boldsymbol{\xi}_t = \pi_\phi(\mathbf{h}_{t-1})$ alone would yield little diversity in the observations. To expose the posterior and summary networks to more varied data initially, we treat the design prior $p(\boldsymbol{\xi})$ as a generic source of exploratory designs and combine it with the learned policy at every rollout step according to

$$\boldsymbol{\xi}_t = \begin{cases} \pi_\phi(\mathbf{h}_{t-1}) & \text{with probability } 1 - \rho_n, \\ \boldsymbol{\xi}_t \sim p(\boldsymbol{\xi}) & \text{with probability } \rho_n, \end{cases} \quad (19)$$

where $\rho_n \in [0, 1]$ is a scheduled exploration probability that is typically decreased over the course of training. The appropriate regime depends on how informative random designs are for the task. If random designs already produce useful observations, an initial phase of pure prior sampling ($\rho_n = 1$) can be used to pretrain the summary and posterior networks before switching to fully policy-driven rollouts ($\rho_n = 0$) (Section 5.1). If random designs are only partially informative but broad early coverage is still useful, an annealed schedule from $\rho_n = 1$ to $\rho_n = 0$ provides a compromise between exploration and policy learning (Section 5.3). If large parts of the design space are uninformative, random-design pretraining can be ineffective or unstable; in this case, starting directly with policy-driven joint training ($\rho_n = 0$) is preferable (Section 5.2). When little prior information about the task is available, these regimes provide a practical order of preference: prior pretraining, annealed mix-in, and direct joint training. Concrete choices of $\rho_n$ for each experiment are described in Section 5, with additional ablations in Appendix B.

## 4. Related Work

**Amortized simulation-based inference.** Amortized SBI methods typically train a global posterior functional, $\mathbf{x} \mapsto q_\psi(\boldsymbol{\theta} \mid \mathbf{x})$ for a *fixed* design $\boldsymbol{\xi}$. The functional can be realized by any generative model, such as normalizing flows (Ardizzone et al., 2019), flow matching (Wildberger et al., 2023), diffusion (Sharrock et al., 2024), or consistency models (Schmitt et al., 2024). A common architectural choice is to separate the model into a summary network $\eta_\omega(\mathbf{x})$, which embeds observation sequences into a fixed-dimensional representation, and an inference network (i.e., a generative

backbone) which can sample from $q_\psi(\boldsymbol{\theta} \mid \eta_\omega(\mathbf{x}))$ (Radev et al., 2020; Chen et al., 2021; 2023). Our method uses design-aware summary networks combined with flexible diffusion-based inference backbones (Arruda et al., 2025).

**Variational Bayesian experimental design.** Neural-based variational formulations replace unknown densities (posterior, marginal, and/or likelihood) with flexible parametric approximations (Foster et al., 2019). When the posterior $p(\boldsymbol{\theta} \mid \mathbf{x}, \boldsymbol{\xi})$ is replaced by a variational approximation $q_\psi(\boldsymbol{\theta} \mid \mathbf{x}, \boldsymbol{\xi})$ in (5), this yields the Barber-Agakov lower bound on the EIG (Barber & Agakov, 2003). More recent work has employed flexible neural models, such as conditional normalizing flows, to parameterize the posterior (Orozco et al., 2024; Dong et al., 2025) or likelihood (Zaballa & Hui, 2025). In all these approaches, however, these variational approximations are used primarily as a surrogate for efficient, differentiable EIG estimation in static (i.e. non-adaptive) design problems, rather than as final amortized artifacts.

**Bayesian adaptive design.** Traditionally, BAD has been formulated as a greedy two-step sequential procedure. At each experiment iteration $t$, one first optimizes the incremental, one-step ahead EIG (6) using gradient-free (von Kügelgen et al., 2019; Hamada et al., 2001; Price et al., 2018) or, more recently, gradient-based surrogates (Foster et al., 2020; Kleinegesse & Gutmann, 2020). Most recently, Iollo et al. (2025) extended this gradient-based strategy to high-dimensional tasks using diffusion models, leveraging a pooled posterior proxy to estimate the *gradients* of the EIG. After observing the experimental outcome, a separate Bayesian inference step is performed, typically using (asymptotically) exact methods such as MCMC or SMC (Drovandi et al., 2014; Kuck et al., 2006; Vincent & Rainforth, 2017), and resorting to simulation-based methods only when the likelihood is intractable (Huan & Marzouk, 2013; Lintusaari et al., 2017; Sisson et al., 2018). This pipeline blueprint separates design optimization from posterior inference, but has to solve the inference problem from scratch at each decision step.

**Amortized Bayesian adaptive design.** The idea of fully amortizing the adaptive design process is to avoid intermediate posterior calculations by directly mapping past experimental data to future design decisions. Foster et al. (2021) were the first to derive the TEIG objective and, using a lower bound on its likelihood-based form (8), train an amortized Deep Adaptive Design (DAD) policy network. The idea has subsequently been extended to differentiable implicit models (Ivanova et al., 2021), to objectives that directly target downstream decision-making utilities (Huang et al., 2024), to semi-amortized settings that introduce local policy updates (Hedman et al., 2025), and to reinforcement learn-

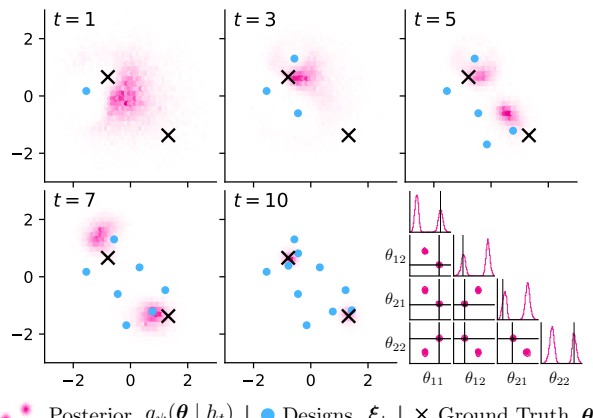

*Figure 2.* Rollout process for Location Finding. *Panels:* posterior samples and chosen designs over time $t$, with crosses marking the true source locations. The second posterior mode is typically uncovered around $t = 10$ measurements. *Bottom right:* corner plot of the learned posterior over the two sources at $t = 10$ shows nearly identical densities at $(\theta_{11}, \theta_{12})$ and $(\theta_{21}, \theta_{22})$, indicating that the model correctly captures exchangeability of the two modes, that is, $p([\theta_{11}, \theta_{12}], [\theta_{21}, \theta_{22}] \mid \mathbf{h}_{10}) = p([\theta_{21}, \theta_{22}], [\theta_{11}, \theta_{12}] \mid \mathbf{h}_{10})$.

ing (RL) based approaches suitable for non-differentiable simulators (Blau et al., 2022; Lim et al., 2022). All of these approaches optimize the design policy in isolation, often leaving accurate posterior estimation as a post-hoc task.

**Unified amortized design and inference.** A recent wave of methods aims to jointly amortize adaptive design and Bayesian inference. Three closely related approaches: RL-sCEE (Blau et al., 2023), vsOED (Shen et al., 2025), and ALINE (Huang et al., 2026) optimize a BA-style variational lower bounds on the TEIG (12) by *explicitly casting the design problem within an RL framework*. All three methods consequently rely on high-variance REINFORCE estimators (in the case of ALINE), or actor-critic algorithms that require training of additional value networks (in the case of RL-sCEE and vsOED).

Furthermore, their dependence on explicit density estimators of the posterior $q_\psi$ necessitates estimators with tractable likelihoods, restricting them to architectures such as normalizing flows or the much less expressive GMMs. In contrast, JADAI frames the problem as optimization over sampled rollouts, demonstrating that the heavy RL machinery is not strictly necessary for effective amortized design and inference. Finally, unlike prior methods, JADAI incorporates implicit generative models (e.g., diffusion models) that afford scalable and flexible inference.

## 5. Experiments

We evaluate our method on three benchmarks that illustrate a progression in both posterior and policy complexity.

*Table 1.* sPCE lower bound on total EIG (↑) for Location Finding (LF) and constant elasticity of substitution (CES) benchmarks. For LF, our posterior-based policies ($u_{10}, u_{20}, u_{30}$) exceed prior baselines for all cases where $T > 10$. For CES, our method outperforms all existing baselines at the standard evaluation horizon $T = 10$. Policies trained with the longest terminal horizon perform best, also at intermediate rollout lengths, e.g., $u_{30}$ vs. $u_{20}$ evaluated at $T = 20$. $L$ is the number of contrastive samples. All values were taken as reported in the corresponding references, and missing entries "−" indicate settings not reported there.

| Method | Location Finding | | | | CES |
|---|---|---|---|---|---|
| | $10T\,2K\,5\cdot10^5L$ | $20T\,2K\,5\cdot10^5L$ | $30T\,2K\,10^6L$ | $30T\,1K\,10^6L$ | $10T\,3K\,10^7L$ |
| Random | $4.79 \pm 0.04$ | $7.00 \pm 0.03$ | $8.30 \pm 0.04$ | $5.17 \pm 0.05$ | $9.05 \pm 0.26$ |
| SG-BOED (Foster et al., 2020) | $5.55 \pm 0.03$ | $7.70 \pm 0.03$ | $8.84 \pm 0.04$ | $5.25 \pm 0.22$ | $9.40 \pm 0.27$ |
| iDAD (Ivanova et al., 2021) | $7.75 \pm 0.04$ | $10.08 \pm 0.03$ | − | − | − |
| DAD (Foster et al., 2021) | $\mathbf{7.97 \pm 0.03}$ | $10.42 \pm 0.03$ | $10.97 \pm 0.04$ | $7.33 \pm 0.06$ | $10.77 \pm 0.15$ |
| RL-BOED (Blau et al., 2022) | − | − | $11.73 \pm 0.04$ | $7.70 \pm 0.06$ | $14.60 \pm 0.10$ |
| RL-sCEE (Blau et al., 2023) | − | − | $12.31 \pm 0.06$ | − | − |
| ALINE (Huang et al., 2026) | − | − | − | $8.91 \pm 0.04$ | $14.37 \pm 0.08$ |
| Ours $u_{10}$ | $6.47 \pm 0.04$ | − | − | − | $\mathbf{14.76 \pm 0.05}$ |
| Ours $u_{20}$ | $6.71 \pm 0.04$ | $\mathbf{10.48 \pm 0.04}$ | − | − | − |
| Ours $u_{30}$ | $6.74 \pm 0.04$ | $\mathbf{10.90 \pm 0.03}$ | $\mathbf{12.82 \pm 0.03}$ | $\mathbf{9.62 \pm 0.02}$ | $\mathbf{14.85 \pm 0.05}$ |

The first two, Location Finding (LF) and Constant Elasticity of Substitution (CES), are standard benchmarks: LF requires only a simple policy but yields a multimodal posterior, whereas CES typically leads to a simple, approximately unimodal posterior but requires a more complex policy. Recently, Iollo et al. (2025) proposed the MNIST Image Discovery (ID) task, which combines both challenges in a high-dimensional observation space and requires a sophisticated policy together with a flexible multimodal posterior.

For LF and CES, we assess policy quality using the sequential prior contrastive estimation (sPCE) (Foster et al., 2021) lower bound on the total expected information gain, while for ID we report the Structural Similarity Index Measure (SSIM) (Wang et al., 2004) and the normalized root-mean-square error (NRMSE) (see Appendix C for details on experiments and metrics).

## 5.1. Location finding

We first consider the Location Finding benchmark of Foster et al. (2021), where the goal is to infer the locations $\boldsymbol{\theta}$ of $K = 1$ or $K = 2$ signal-emitting sources (depending on the experimental setting) from noisy measurements of their summed intensity $\mathbf{x}$ at adaptively chosen measurement positions $\boldsymbol{\xi}$ (experimental details in Section C.2). Because the optimal policy is relatively simple, we pre-train the summary and posterior networks under a random design policy before joint training, as discussed in Section 3.4.

Qualitatively, the learned policy balances exploration and exploitation: during the initial rollout steps, when posterior uncertainty is high, it explores the design space broadly; as uncertainty decreases, it concentrates measurements in regions of high posterior density, placing most posterior mass close to the true source locations while continuing to explore until both sources (i.e., $K = 2$) have been identified

(see Figure 2 for a typical rollout). Since the policy only observes the current summary state $\mathbf{h}_t$, this behavior indicates that $\mathbf{h}_t$ encodes which regions have already been probed and where posterior mass is concentrated. Furthermore, the corner plot in Figure 2 shows nearly identical densities at $(\theta_{11}, \theta_{12})$ and $(\theta_{21}, \theta_{22})$, confirming that $q_\psi(\boldsymbol{\theta} \mid \mathbf{h}_t)$ learns the full joint posterior and respects exchangeability of the two sources.

Quantitatively, we evaluate the policy using the sPCE lower bound on the total EIG. Our policies are competitive across all settings and outperform prior approaches for both $K = 1$ and $K = 2$ sources whenever $T > 10$ (Table 1). Moreover, training with a longer terminal horizon further improves performance at shorter evaluation horizons: policies trained with $T = 30$ achieve higher sPCE than those trained with $T = 20$ when both are evaluated at $T = 20$. Since the same summary and policy networks are applied at every time step, optimizing per-step posterior losses up to $T = 30$ includes all losses for $t \leq 20$ and additionally trains the networks on later, typically more concentrated posteriors. This extra training signal can refine how the summary network and the policy respond to similar configurations that already occur earlier in the rollout, leading to summary representations that generalize better across rollout lengths.

Additional ablation and posterior quality results (Appendix B) and examples for a longer terminal horizon ($T = 30$, Figure 5) are presented in the appendix.

## 5.2. Constant elasticity of substitutions

As a more challenging design problem, we consider the Constant Elasticity of Substitution (CES) (Arrow et al., 1961; Foster et al., 2019) benchmark next, where an agent rates the difference in subjective utility $\mathbf{x}$ between a pair of two baskets $\boldsymbol{\xi}$ each with $K = 3$ goods, and the goal is to in-

fer a five-dimensional preference parameter $\boldsymbol{\theta}$. Informative designs lie in a narrow "sweet spot" between nearly identical baskets (indifference) and very different baskets, where noise and sigmoid saturation dominate (Foster et al., 2019). In practice, this makes random designs largely uninformative: random-policy pretraining led to unstable training or collapsed weights, so for CES we train summary, policy, and posterior jointly from the beginning (see Section C.3).

Our method outperforms all state-of-the-art approaches at the commonly used evaluation horizon $T = 10$ (see Table 1). As in Location Finding, training with a longer terminal horizon yields additional gains: policies trained with $T = 30$ achieve slightly higher sPCE when evaluated at $T = 10$ than policies trained directly with $T = 10$.

### 5.3. Image Discovery

Finally, we evaluate our method on the high-dimensional MNIST Image Discovery task introduced by "CoDiff" (Iollo et al., 2025). At each step, the policy selects a spatial location $\boldsymbol{\xi}$, and the simulator reveals a local measurement patch $\mathbf{x}$. The downstream task is to reconstruct the full digit image $\boldsymbol{\theta}$ from a sequence of such measurements. We follow CoDiff's simulator implementation but also consider variants with additive measurement noise ($\sigma > 0$) that remove useful signal outside the measurement mask (see Section C.4).

During training, we follow the mixed-policy scheme from Section 3.4, starting with mostly random designs and gradually annealing the probability $\rho_n$ of random actions to zero so that, over time, rollouts are generated entirely by the learned policy. Alongside the diffusion-based posterior estimator, we also train a flow matching variant with the same architecture (see Section A.3 for details), demonstrating that our framework can incorporate score- and flow matching-based objectives equally well.

Intuitively, an effective policy should first gather information that disambiguates the digit class and then refine the digit's shape. Qualitatively, our learned policies exhibit this behavior: early measurements are placed on non-overlapping, class-discriminative regions, while later measurements refine local details (see Figure 3, and Figure 7 for additional examples). The posterior typically converges to the correct digit shape in fewer than $T = 6$ measurements.

Quantitatively, we average SSIM and NRMSE results over 30 posterior samples for the whole validation split at each rollout step (Figure 4), achieving the best results across all noise levels (Table 2). Both metrics improve rapidly during the first few measurements, indicating that most information is gained early, with later steps primarily refining the reconstruction. Notably, under a random policy, CoDiff and our posterior network achieve similar SSIM values, indicating that the performance gap is primarily due to the policy rather

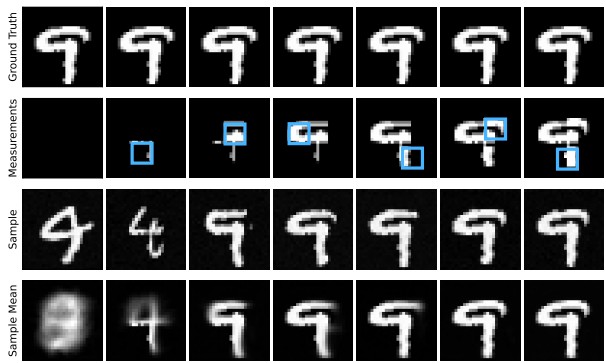

*Figure 3.* Image discovery rollout. Each column is one measurement step. From *top* to *bottom*: ground-truth digit, cumulative measurements with the newly chosen design highlighted in blue, one posterior sample, and the posterior mean over 100 samples. The correct digit class is typically identified after 1–2 measurements, after which the policy mainly refines local structure in uncertain regions.

*Table 2.* SSIM ($\uparrow$) and Posterior NRMSE ($\downarrow$, not reported in CoDiff (Iollo et al., 2025) and marked as "$-$") at the terminal horizon ($T = 6$) on the MNIST validation set. Our methods outperform CoDiff, with Flow Matching (FM) and Diffusion Models (DM) achieving similar performance, and little degradation from additive measurement noise ($\sigma$).

| $\sigma$ | Method | SSIM $\uparrow$ | NRMSE $\downarrow$ |
|---|---|---|---|
| 0 | CoDiff (random) | 0.463 | $-$ |
| | Ours (random, DM) | $0.478 \pm 0.002$ | $2.056 \pm 0.007$ |
| | Ours (random, FM) | $0.451 \pm 0.001$ | $2.168 \pm 0.007$ |
| | CoDiff | 0.826 | $-$ |
| | Ours (DM) | $0.968 \pm 0.001$ | $\mathbf{0.177 \pm 0.001}$ |
| | Ours (FM) | $\mathbf{0.988 \pm 0.001}$ | $0.185 \pm 0.002$ |
| 0.001 | Ours (DM) | $0.966 \pm 0.001$ | $\mathbf{0.208 \pm 0.002}$ |
| | Ours (FM) | $\mathbf{0.985 \pm 0.001}$ | $0.218 \pm 0.002$ |
| 0.01 | Ours (DM) | $0.960 \pm 0.001$ | $\mathbf{0.246 \pm 0.002}$ |
| | Ours (FM) | $\mathbf{0.981 \pm 0.001}$ | $0.249 \pm 0.002$ |

than differences in sampling or network architectures.

## 6. Conclusion

We introduce JADAI, a new framework for jointly amortizing adaptive experimental design and posterior inference via an incremental posterior loss as a proxy for the classical TEIG. Our method enables posterior estimation at any step of the sequential design process, rather than only after a fixed horizon, and thus connects naturally to active data-acquisition use cases. At test time, experts may override or modify the designs proposed by the learned policy while being informed by intermediate approximate posteriors.

In the default setting (without user intervention), full rollouts run in milliseconds, which is comparable to recent methods (Table 2; Huang et al., 2026) on low-dimensional

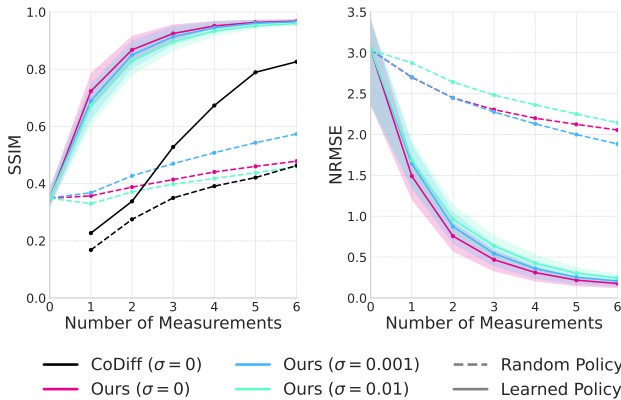

*Figure 4.* Validation SSIM ($\uparrow$) and NRMSE ($\downarrow$) as a function of the number of measurements for CoDiff and our methods (using diffusion models). Shaded bands indicate the interquartile range over the validation set. Our learned methods achieve better SSIM and NRMSE than CoDiff and the random baselines at all steps and remain robust to additive measurement noise.

tasks and roughly an order of magnitude faster on our high-dimensional benchmark. Posterior sampling, however, remains a bottleneck: generating 10,000 samples takes a few seconds in the high-dimensional case, making sub-second deployment challenging. A promising direction is to distill the posterior estimator post hoc into a faster surrogate.

Both qualitatively and quantitatively, JADAI typically matches or improves upon prior work, particularly on high-dimensional inference problems, while approximating multimodal posteriors and maintaining effective policies for more complex design choices such as CES. A natural direction for future work is to investigate the limits of this approach as the design space becomes increasingly complex, for instance, by considering higher-dimensional designs like spatial patterns or time-series stimuli.

Although JADAI is applicable when the likelihood is not available in closed form, our experiments rely on differentiating through the simulator. As differentiable simulators in autodiff frameworks become more common (Lavin et al., 2022; Filipovich & Lvovsky, 2024; Stoffel et al., 2025; Deb et al., 2025), this setting is increasingly relevant; however, extending JADAI to purely black-box simulators remains an important direction and will likely require gradient-free design optimization.

In this work, we used separate networks for the policy, summary, and posterior. By contrast, the success of ALINE (Huang et al., 2026) and related work (Huang et al., 2024; Zhang et al., 2025; Chang et al., 2025) stems in part from a well-chosen transformer backbone with multiple task-specific heads. Thus, a natural extension of JADAI would be to keep the posterior network separate but let the policy and summary networks share a transformer encoder.

## Acknowledgements

This work is partially funded by the Deutsche Forschungs-gemeinschaft (DFG, German Research Foundation) Project 528702768 and Collaborative Research Center 391 (Spatio-Temporal Statistics for the Transition of Energy and Transport) – 520388526. STR and NB are supported by the National Science Foundation under Grant No. 2448380. We thank Jerry M. Huang for his support in creating Figure 1.

## Impact Statement

We propose a unified framework that accelerates and scales Bayesian adaptive design and posterior inference. By using diffusion-based models, our approach can extend adaptive design to high-dimensional and multimodal problems where classical design-and-infer pipelines are computationally impractical. There are many potential societal consequences of this line of work, none of which we feel must be specifically highlighted here.

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

## A. Method Details

### A.1. Relation between JADAI's objective and the Barber-Agakov bound

**BA per horizon** The sequential cross-entropy estimator (sCEE) introduced in (Blau et al., 2023) can be viewed as the posterior-form TEIG bound corresponding to Eq. (9), see also (Foster et al., 2021). The same construction applies at any $t \leq T$. The TEIG at rollout step $t$ is

$$\text{TEIG}_t := \mathbb{E}_{p(\boldsymbol{\theta}, \mathbf{h}_t)} \left[ \log \frac{p(\boldsymbol{\theta} \mid \mathbf{h}_t)}{p(\boldsymbol{\theta})} \right]. \tag{20}$$

Following (Theorem 1, Corollary 2; Blau et al., 2023), replacing $p(\boldsymbol{\theta} \mid \mathbf{h}_t)$ by any approximate posterior $q_\psi(\boldsymbol{\theta} \mid \mathbf{h}_t)$ yields

$$\text{TEIG}_t = \underbrace{\mathbb{E}_{p(\boldsymbol{\theta}, \mathbf{h}_t)} \left[ \log \frac{q_\psi(\boldsymbol{\theta} \mid \mathbf{h}_t)}{p(\boldsymbol{\theta})} \right]}_{\text{sCEE}_t(\psi)} + \delta_t(\psi), \qquad \delta_t(\psi) := \mathbb{E}_{p(\mathbf{h}_t)} \left[ \text{KL} \left( p\left(\boldsymbol{\theta} \mid \mathbf{h}_t\right) \| q_\psi \left(\boldsymbol{\theta} \mid \mathbf{h}_t\right) \right) \right] \geq 0. \tag{21}$$

Thus, $\text{sCEE}_t(\psi) \leq \text{TEIG}_t$ with equality iff $q_\psi = p(\boldsymbol{\theta} \mid \mathbf{h}_t)$. This inequality is agnostic to how $q_\psi$ was obtained; it depends only on the resulting posterior approximation.

The gap $\delta_t(\psi)$ comprises the standard approximation error of a finite-capacity neural network, present for any posterior model including normalizing flows, as well as an additional component when $q_\psi$ is trained via score estimation ($\mathcal{L}_{\text{post}}^{\text{diff}}$, Eq. (33)) or flow matching ($\mathcal{L}_{\text{post}}^{\text{fm}}$, Eq. (40)), rather than direct density optimization (Song et al., 2021a;b; Lu et al., 2022). Both components can be reduced over the course of training and, in all cases, $\text{sCEE}_t(\psi) \leq \text{TEIG}_t$ remains valid regardless of the magnitude of $\delta_t(\psi)$.

**Summing TEIGs** Summing Eq. (21) over $t = 0, \ldots, T$ gives

$$\sum_{t=0}^{T} \text{sCEE}_t(\psi) = \sum_{t=0}^{T} \text{TEIG}_t - \sum_{t=0}^{T} \delta_t(\psi) \leq \sum_{t=0}^{T} \text{TEIG}_t. \tag{22}$$

Since $\text{TEIG}_t$ does not depend on $\psi$, maximizing $\sum_t \text{sCEE}_t(\psi)$ with respect to $\psi$ is equivalent to minimizing $\sum_t \delta_t(\psi)$. The detached telescoping provides exactly this multi-horizon optimization. To see this, we can define an alternative objective

$$\mathcal{L}^{\text{sum}}(\psi) := \sum_{t=0}^{T} \mathbb{E}_{p(\boldsymbol{\theta}, \mathbf{h}_t)} \left[ \ell_t \right], \tag{23}$$

where $\ell_t = \mathcal{L}_{\text{post}}(\boldsymbol{\theta}, \mathbf{h}_t)$ is the per-step posterior loss (Eq. (11)). Recall from Eq. (16) that the detached utility $u_T = \sum_{t=0}^{T} \left( \ell_{t-1}^* - \ell_t \right)$ with $\ell_{-1} := 0$ telescopes to $u_T = -\ell_T$ as a scalar, but the gradient with respect to $\psi$ does not telescope due to the stop-gradient in (Eq. (18)):

$$\nabla_\psi \mathcal{L}(\psi) = \sum_{t=0}^{T} \nabla_\psi \mathbb{E} \left[ \ell_t \right] = \nabla_\psi \mathcal{L}^{\text{sum}}(\psi). \tag{24}$$

The stop-gradient thus ensures that $q_\psi$ receives posterior-training signal from every rollout step $t = 0, \ldots, T$ (not only the terminal signal) independently (i.e., no nested BPTT due to the optimization along the rollout), even though the scalar objective $\mathcal{L} = \mathbb{E}[\ell_T]$ involves only the terminal loss.

In the density case ($\ell_t = \ell_t^{\text{flow}} = -\log q_\psi(\boldsymbol{\theta} \mid \mathbf{h}_t)$), the alternative objective $\mathcal{L}^{\text{sum}}(\psi)$ reduces to $\mathcal{L}^{\text{sum}}(\psi) = \sum_{t=0}^{T} \mathbb{E} \left[ -\log q_\psi(\boldsymbol{\theta} \mid \mathbf{h}_t) \right]$. Since no $\text{TEIG}_t$ depends on $\psi$, it follows from Eq. (22) that

$$\nabla_\psi \mathcal{L}^{\text{sum}}(\psi) = \sum_{t=0}^{T} \nabla_\psi \delta_t(\psi), \tag{25}$$

detaching the gradients in Eq. (24) exactly minimizes the summed BA gaps $\sum_t \delta_t(\psi)$.

In contrast, in the score-based case, $\ell_t$ is instantiated as $\mathcal{L}_{\text{post}}^{\text{diff}}$ from Eq. (33) or $\mathcal{L}_{\text{post}}^{\text{fm}}$ from Eq. (40), rather than as the density-based objective $-\log q_\psi(\boldsymbol{\theta} \mid \mathbf{h}_t)$. As noted in Section 3.2, the Barber-Agakov interpretation as an explicit log-density-ratio optimization no longer holds directly, so the exact identity in Eq. (25) does not apply. However, the gradient formulation in Eq. (24) remains unchanged, i.e. the detached telescoping still accumulates $\nabla_\psi \ell_t$ across all $t \leq T$, serving as a surrogate for minimizing the summed BA gaps $\sum_t \delta_t(\psi)$, where each gradient step reduces $\delta_t(\psi)$ and tightens $\text{sCEE}_t(\psi)$ toward $\text{TEIG}_t$.

## A.2. Diffusion model background

In the following, we provide a brief overview of our diffusion model (Ho et al., 2020; Song & Ermon, 2019; Kingma & Gao, 2023) used to approximate the posterior $p(\boldsymbol{\theta} \mid \cdot) \approx q_\psi(\boldsymbol{\theta} \mid \cdot)$. More details on diffusion models in a general setting can be found in (Karras et al., 2022) and, for simulation-based inference, in (Arruda et al., 2025).

A diffusion model learns how to gradually denoise a sample from a base distribution $\mathbf{z}_1 \sim p(\mathbf{z}_1) = \mathcal{N}(\mathbf{0}, \mathbf{I})$, typically a standard Gaussian, towards the target data distribution $\boldsymbol{\theta} \equiv \mathbf{z}_0 \sim p(\mathbf{z}_0 \mid \cdot)$. At the core of this learning process is the forward corruption process

$$\mathrm{d}\mathbf{z}_\tau = f(\tau)\,\mathbf{z}_\tau\,\mathrm{d}\tau + g(\tau)\,\mathrm{d}\mathrm{W}_\tau, \tag{26}$$

where $f(\tau)$ and $g(\tau)$ are drift and diffusion coefficients respectively and $\mathrm{dW}$ defines a Wiener process. Starting from $\tau = 0$, this forward process gradually adds Gaussian noise to a sample from the target distribution until it approximately follows the base distribution at $\tau = 1$.

This construction allows computing the marginal densities $p(\mathbf{z}_\tau \mid \mathbf{z}_0)$ analytically for every $0 < \tau < 1$:

$$p(\mathbf{z}_\tau \mid \mathbf{z}_0) = \mathcal{N}(\mathbf{z}_\tau \mid \alpha_\tau\,\mathbf{z}_0, \sigma_\tau^2) \quad \Longleftrightarrow \quad \mathbf{z}_\tau = \alpha_\tau\,\mathbf{z}_0 + \sigma_\tau\,\boldsymbol{\epsilon} \quad \text{with} \quad \boldsymbol{\epsilon} \sim \mathcal{N}(\mathbf{0}, \mathbf{I}). \tag{27}$$

The reverse SDE has the form

$$\mathrm{d}\mathbf{z}_\tau = \tilde{f}(\tau, \mathbf{z}_\tau)\,\mathrm{d}\tilde{\tau} + g(\tau)\,\mathrm{d}\mathrm{W}_\tau \quad \text{with} \quad \tilde{f}(\tau, \mathbf{z}_\tau) = f(\tau)\,\mathbf{z}_\tau - g(\tau)^2\,\nabla_{\mathbf{z}_\tau} \log p(\mathbf{z}_\tau \mid \mathbf{z}_0), \tag{28}$$

where $\tilde{f}$ is a new drift term depending on the score $\nabla_{\mathbf{z}_\tau} \log p(\mathbf{z}_\tau \mid \mathbf{z}_0)$ of the conditional distribution from the forward process (27). A neural network is trained to approximate that score via a weighted score-matching objective:

$$s_\psi = \underset{\psi}{\arg\min}\ \mathbb{E}_{\mathbf{z}_0,\,\mathbf{x} \sim p(\mathbf{z}_0, \mathbf{x}),\,\tau \sim \mathcal{U}(0,1),\,\boldsymbol{\epsilon} \sim \mathcal{N}(\mathbf{0}, \mathbf{I})} \left[ w_\tau \left\| s_\psi(\mathbf{z}_\tau, \mathbf{x}, \tau) - \nabla_{\mathbf{z}_\tau} \log p(\mathbf{z}_\tau \mid \mathbf{z}_0) \right\|_2^2 \right], \tag{29}$$

where $\mathbf{x}$ denotes an optional condition variable, $w_\tau$ a diffusion time-dependent weighting and $\mathbf{z}_\tau$ computed as in (27) with coefficients defined in the following.

The coefficients in the marginal density (27) define a noise schedule. They control how much noise is added at each step $\tau \in [0, 1]$ and are related to the SDE coefficients in (26) via $f(\tau) = \alpha_\tau'/\alpha_\tau$ and $g(\tau)^2 = 2\sigma_\tau(\sigma_\tau' - \alpha_\tau'/\alpha_\tau\sigma_\tau)$. In our experiments, we chose a *variance-preserving* schedule such that the relation between these two is given by $1 = \alpha_\tau^2 + \sigma_\tau^2$. Although diffusion time $\tau$ is sampled uniformly in (29) and controls both coefficients, the noise schedule is often parameterized in terms of the log signal-to-noise ratio $\lambda(\tau) = \log(\alpha_\tau^2/\sigma_\tau^2)$, which allows shifting emphasis towards specific regions of the noise spectrum. For all experiments, we used a *cosine schedule* $\lambda(\tau) = -2\log\left(\tan\frac{\pi\tau}{2}\right)$ that places more probability mass near intermediate SNR levels ($\lambda_\tau \approx 0$) than the original linear schedule.

Instead of directly predicting the conditional score, one can predict the noise $\boldsymbol{\epsilon}$ or an interpolation between data and noise $\mathbf{v}$ at each time step $\tau$ and replace the score $s$ in (29) accordingly. The relation between noise and the score is:

$$s = \nabla_{\mathbf{z}_\tau} \log p(\mathbf{z}_\tau \mid \mathbf{z}_0) = -\frac{\boldsymbol{\epsilon}}{\sigma_\tau}, \tag{30}$$

so predicting $\boldsymbol{\epsilon}$ is equivalent to predicting the score up to a known scaling factor. The noise target is simply $\boldsymbol{\epsilon} \sim \mathcal{N}(\mathbf{0}, \mathbf{I})$ while the target for $\mathbf{v}$-*prediction* is defined as the interpolation between data and noise, which in the variance-preserving case is:

$$\mathbf{v}_\tau = \alpha_\tau\boldsymbol{\epsilon} - \sigma_\tau\mathbf{z}_0 \quad \Longleftrightarrow \quad \boldsymbol{\epsilon} = \sigma_\tau\mathbf{z}_\tau + \alpha_\tau\mathbf{v}_\tau \tag{31}$$

In our case, the network is parameterized for $\mathbf{v}$-prediction, and its outputs are converted to the noise domain via (31) so that the chosen noise and weighting schedules remain unchanged by the parameterization.

After training, we draw approximate posterior samples $\boldsymbol{\theta} \sim q_\psi(\boldsymbol{\theta} \mid \mathbf{h}_t)$ by starting from a base sample $\mathbf{z}_1 \sim \mathcal{N}(\mathbf{0}, \mathbf{I})$ and solving the probability-flow ODE associated with the reverse SDE in Eq. (26). Concretely, we use the *deterministic ODE*

$$\frac{\mathrm{d}\mathbf{z}_\tau}{\mathrm{d}\tau} = f(\tau)\,\mathbf{z}_\tau - \frac{1}{2}g(\tau)^2\,s_\psi(\mathbf{z}_\tau, \mathbf{h}_t, \tau), \tag{32}$$

where $s_\psi(\mathbf{z}_\tau, \mathbf{h}_t, \tau)$ denotes the learned conditional score. In practice, the network predicts the velocity $\mathbf{v}_\tau$, which is converted to a score estimate by using the relations between velocity, noise, and score from (31) and (30). We integrate this ODE numerically from $\tau = 1$ to $\tau = 0$ using an *explicit Euler* solver with $N = 1000$ equidistant steps.

**Posterior diffusion loss in our setting.** In our SBI setting, we use a diffusion model to approximate the conditional distribution $p(\boldsymbol{\theta} \mid \mathbf{h}_t)$ at each step $t$, with $\boldsymbol{\theta} \equiv \mathbf{z}_0$ and the summary $\mathbf{h}_t \equiv \mathbf{x}$ playing the role of the conditioning variable. For each training iteration, we sample $\tau \sim \mathcal{U}(0, 1)$ and $\boldsymbol{\epsilon} \sim \mathcal{N}(\mathbf{0}, \mathbf{I})$ once, construct $\mathbf{z}_\tau$ from (27) and reuse the same $(\tau, \boldsymbol{\epsilon})$ at all rollout steps along the trajectory. The network takes $(\mathbf{z}_\tau, \mathbf{h}_t, \tau)$ as input and predicts $\mathbf{v}_\tau$, which we convert to the noise domain to obtain $\boldsymbol{\epsilon}_\psi$ using (31). For a fixed sampled pair $(\tau, \boldsymbol{\epsilon})$, the resulting per-step diffusion posterior loss is

$$\mathcal{L}_{\text{post}}^{\text{diff}}(\boldsymbol{\theta}, \mathbf{h}_t; \phi, \omega, \psi, \tau, \boldsymbol{\epsilon}) := w_\tau \left\| \boldsymbol{\epsilon}_\psi(\mathbf{z}_\tau, \mathbf{h}_t, \tau; \pi_\phi, \eta_\omega) - \boldsymbol{\epsilon} \right\|_2^2, \tag{33}$$

where $\mathbf{h}_t$ depends on $(\phi, \omega)$ implicitly through the policy and summary networks, $\mathbf{h}_t = \eta_\omega(\{(\boldsymbol{\xi}_k, \mathbf{x}_k)\}_{k=1}^t)$ and $\boldsymbol{\xi}_k = \pi_\phi(\mathbf{h}_{k-1})$. In the notation of the main text, where the generic per-step posterior loss is $\ell_t(\boldsymbol{\theta}, \mathbf{h}_t) := \mathcal{L}_{\text{post}}(\boldsymbol{\theta}, \mathbf{h}_t; \phi, \omega, \psi)$, we simply set

$$\ell_t(\boldsymbol{\theta}, \mathbf{h}_t) = \mathcal{L}_{\text{post}}^{\text{diff}}(\boldsymbol{\theta}, \mathbf{h}_t; \phi, \omega, \psi, \tau, \boldsymbol{\epsilon}) \tag{34}$$

for the diffusion-based posterior estimator. Plugging this choice of $\ell_t$ into the utility $u_T$ in Eq. (16) and the global objective $\mathcal{L}$ in Eq. (17) yields the diffusion-specific training objective used in our experiments:

$$\mathcal{L}^{\text{diff}}(\phi, \psi, \omega) = \mathbb{E}_{\boldsymbol{\theta} \sim p(\boldsymbol{\theta}),\, h_{0:T} \sim p(h_{0:T} \mid \boldsymbol{\theta}, \pi_\phi, \eta_\omega),\, \tau \sim \mathcal{U}(0,1),\, \boldsymbol{\epsilon} \sim \mathcal{N}(\mathbf{0}, \mathbf{I})} \left[ -u_T^{\text{diff}}(\boldsymbol{\theta}, h_{0:T}, \tau, \boldsymbol{\epsilon}) \right]. \tag{35}$$

In practice, the joint expectation in $\mathcal{L}^{\text{diff}}$ is approximated via Monte Carlo over quadruples $(\boldsymbol{\theta}, h_{0:T}, \tau, \boldsymbol{\epsilon})$. Each training iteration draws a minibatch of parameters $\boldsymbol{\theta} \sim p(\boldsymbol{\theta})$, simulates the corresponding rollout summary states $h_{0:T} \sim p(h_{0:T} \mid \boldsymbol{\theta}, \pi_\phi, \eta_\omega)$, and, for each rollout in the minibatch, samples a single pair $(\tau, \boldsymbol{\epsilon})$ with $\tau \sim \mathcal{U}(0, 1)$ and $\boldsymbol{\epsilon} \sim \mathcal{N}(\mathbf{0}, \mathbf{I})$. The same $(\tau, \boldsymbol{\epsilon})$ is reused across all steps $t$ of that trajectory when evaluating $\mathcal{L}_{\text{post}}^{\text{diff}}(\boldsymbol{\theta}, \mathbf{h}_t; \phi, \omega, \psi, \tau, \boldsymbol{\epsilon})$ and aggregating the utility $u_T^{\text{diff}}$ in Eq. (16).

## A.3. Flow matching background

Flow matching provides an alternative to score-based diffusion models by instead of simulating a stochastic forward process, one specifies interpolation paths $\{\mathbf{z}_\tau\}_{\tau \in [0,1]}$ and learns the associated velocity field of the probability-flow ODE directly (Liu et al., 2022; Lipman et al., 2023). Conditional applications in the SBI setting are discussed in Wildberger et al. (2023) and in (Arruda et al., 2025).

As in the diffusion setup above, let $\boldsymbol{\theta} \equiv \mathbf{z}_0 \sim p(\boldsymbol{\theta})$ denote a sample from the target distribution and let $\mathbf{z}_1 \equiv \boldsymbol{\epsilon} \sim \mathcal{N}(\mathbf{0}, \mathbf{I})$ denote Gaussian noise. A simple linear interpolation is

$$\mathbf{z}_\tau = \alpha_\tau \mathbf{z}_0 + \sigma_\tau \boldsymbol{\epsilon}, \qquad \tau \in [0, 1], \tag{36}$$

with the flow-matching schedule $\alpha_\tau = 1 - \tau$ and $\sigma_\tau = \tau$. The associated probability-flow ODE has the velocity field

$$v(\mathbf{z}_\tau, \mathbf{h}_t, \tau) := \frac{\mathrm{d}\mathbf{z}_\tau}{\mathrm{d}\tau} = \frac{\mathrm{d}\alpha_\tau}{\mathrm{d}\tau}\mathbf{z}_0 + \frac{\mathrm{d}\sigma_\tau}{\mathrm{d}\tau}\boldsymbol{\epsilon} = -\mathbf{z}_0 + \boldsymbol{\epsilon} = \boldsymbol{\epsilon} - \boldsymbol{\theta}, \tag{37}$$

which is parameterized by a neural network $v_\psi(\mathbf{z}_\tau, \mathbf{h}_t, \tau) \approx v(\mathbf{z}_\tau, \mathbf{h}_t, \tau)$ and is constant along the path and does not depend on $\tau$ explicitly.

Sampling from the approximate posterior $\boldsymbol{\theta} \sim q_\psi(\boldsymbol{\theta} \mid \mathbf{h}_t)$ can be done by solving the probability-flow ODE

$$\frac{\mathrm{d}\mathbf{z}_\tau}{\mathrm{d}\tau} = v_\psi(\mathbf{z}_\tau, \mathbf{h}_t, \tau), \tag{38}$$

starting from a base sample $\mathbf{z}_1 = \boldsymbol{\epsilon} \sim \mathcal{N}(\mathbf{0}, \mathbf{I})$ at $\tau = 1$ and integrating backwards to $\tau = 0$ with an explicit Euler solver and $N = 1000$ steps:

$$\mathbf{z}_{\tau_{k-1}} = \mathbf{z}_{\tau_k} - \frac{1}{N} v_\psi(\mathbf{z}_{\tau_k}, \mathbf{h}_t, \tau_k), \qquad \tau_k = \frac{k}{N}, \quad k = 1, \dots, N. \tag{39}$$

**Posterior flow-matching loss in our setting.** As in the diffusion case (see Section A.2), we use the flow-matching objective as the per-step posterior loss for the generic training objective in Section 3. For a simulated pair $(\boldsymbol{\theta}, \mathbf{h}_t)$ at design step $t$, and a fixed sampled pair $(\tau, \boldsymbol{\epsilon})$, we define

$$\mathcal{L}_{\text{post}}^{\text{flow}}(\boldsymbol{\theta}, \mathbf{h}_t; \phi, \omega, \psi, \tau, \boldsymbol{\epsilon}) := w_\tau \left\| v_\psi(\mathbf{z}_\tau, \mathbf{h}_t, \tau; \pi_\phi, \eta_\omega) - (\boldsymbol{\epsilon} - \boldsymbol{\theta}) \right\|_2^2, \tag{40}$$

where

$$\mathbf{z}_\tau = (1 - \tau)\,\boldsymbol{\theta} + \tau\,\boldsymbol{\epsilon} \tag{41}$$

is the interpolated state along the flow-matching path (36), and $\mathbf{h}_t$ depends on $(\phi, \omega)$ implicitly through the policy and summary networks, $\mathbf{h}_t = \eta_\omega(\{(\boldsymbol{\xi}_k, \mathbf{x}_k)\}_{k=1}^t)$ and $\boldsymbol{\xi}_k = \pi_\phi(\mathbf{h}_{k-1})$. Comparing with the generic per-step posterior loss $\ell_t(\boldsymbol{\theta}, \mathbf{h}_t) := \mathcal{L}_{\text{post}}(\boldsymbol{\theta}, \mathbf{h}_t; \phi, \omega, \psi)$ from Section 3, the flow-matching instance is

$$\ell_t(\boldsymbol{\theta}, \mathbf{h}_t) \equiv \ell_t^{\text{fm}}(\boldsymbol{\theta}, \mathbf{h}_t; \phi, \omega, \psi, \tau, \boldsymbol{\epsilon}) := \mathcal{L}_{\text{post}}^{\text{fm}}(\boldsymbol{\theta}, \mathbf{h}_t; \phi, \omega, \psi, \tau, \boldsymbol{\epsilon}). \tag{42}$$

Substituting $\ell_t^{\text{fm}}$ for $\ell_t$ in the utility $u_T$ (16) and the population objective (17) yields the flow-matching version of our joint design-and-inference objective, fully analogous to the diffusion-based objective in (35). In practice, we approximate the expectation over $(\boldsymbol{\theta}, h_{0:T}, \tau, \boldsymbol{\epsilon})$ in the same way as for diffusion models (see Section A.2 and Section 3), using minibatches of rollout trajectories and keeping $(\tau, \boldsymbol{\epsilon})$ fixed along each trajectory.

## B. Additional Results

### B.1. Ablations

Here, we present additional ablation results for the location-finding experiment (DAD setting; see Table 5) that investigate several implementation choices introduced in Section 3.4 and summarized in Table 3. Without detaching the summary state before the policy, i.e., when allowing nested backpropagation through time (BPTT), training becomes unstable.

Using a fixed rollout length $r = 30 \equiv T$ equal to the maximum horizon instead of sampling $r \sim \mathcal{U}\{1, \ldots, R(n, T)\}$ at each training iteration leads to better sPCE at the terminal horizon (Q30), but worse performance at intermediate horizons (Q10). Thus, if the maximum number of experiments is known before the study is executed, it is advisable to train the framework with a fixed maximum rollout length. When this maximum is unknown or when active data acquisition is considered, the resulting model from sampling $r$ produces more informative intermediate results during inference.

Even though all experiments considered here are relatively undemanding in terms of memory, also because of the simple architectures we use, we also study truncated BPTT by cutting the gradient computation graph for all observation-design pairs that are not the most recent "$W$" ones (see Algorithm 1). This results in noticeably worse performance than our vanilla configuration, but the model still outperforms the closest competitor at long rollout lengths (see Table 1).

*Table 3. Ablations: sPCE Mean $\pm$ SE over 5 seeds.*

| Ablation | sPCE | |
|---|---|---|
| | Q30 | Q10 |
| Nested BPTT | $7.935 \pm 0.434$ | $5.223 \pm 0.248$ |
| fix $r = 30$ | $13.186 \pm 0.049$ | $6.433 \pm 0.127$ |
| $W = 5$ | $12.536 \pm 0.060$ | $6.533 \pm 0.032$ |
| vanilla | $12.708 \pm 0.081$ | $6.783 \pm 0.019$ |

### B.2. Recommendation on design prior mix-in

We additionally ablated the design-prior mix-in strategy in the same location-finding setting as above. Starting joint training directly with fully policy-driven rollouts ($\rho_n = 0$) achieved a terminal sPCE of $12.348 \pm 0.068$, while an annealed mix-in schedule analogous to the Image Discovery experiment achieved $12.352 \pm 0.044$. Both are close to, but slightly below, the vanilla configuration with design-prior pretraining ($12.708 \pm 0.081$). Thus, for this benchmark, several mix-in regimes lead to comparable performance once training is stable. This supports the practical guidance (Section 3.4): the choice of $\rho_n$ should primarily depend on whether random designs provide informative observations, with prior pretraining being preferable when they do, annealing being useful when broad initial coverage helps, and direct joint training being safer when random designs are mostly uninformative.

## B.3. Posterior quality for Location Finding

To complement the policy-level sPCE evaluation, we additionally evaluate posterior quality in the bimodal Location Finding setting used above. For each evaluation instance, we compute empirical Wasserstein distances (Section C.1) between posterior samples from the learned estimator, high-quality MCMC reference samples, and the corresponding ground-truth parameters (Table 4). We denote the latter by GT. Lower values indicate closer agreement between the compared distributions or point estimates.

As a point of reference, we also evaluate ALINE (Huang et al., 2026), since it is a recent method that also jointly amortizes policy learning and posterior inference. However, ALINE restricts the posterior estimator to a Gaussian mixture model and reports Location Finding results for a unimodal one-source setting. Here, we train and evaluate ALINE on the standard two-source Location Finding benchmark, which induces a multimodal and exchangeable posterior. Thus, this comparison should be interpreted as a stress test of posterior representation quality in the multimodal setting, rather than as a reproduced number from the ALINE paper.

*Table 4. Posterior-quality evaluation for bimodal Location Finding. Empirical Wasserstein distances, reported as mean $\pm$ SE, for the two-source Location Finding setting and 30 rollout steps (Table 5). ALINE is included as a reference joint-amortization method. Lower is better.*

| Method | Comparison | Wasserstein distance $\downarrow$ |
|---|---|---|
| JADAI | $W(\text{JADAI}, \text{MCMC})$ | $0.07 \pm 0.01$ |
| JADAI | $W(\text{JADAI}, \text{GT})$ | $0.09 \pm 0.03$ |
| Reference | $W(\text{MCMC}, \text{GT})$ | $0.11 \pm 0.03$ |
| ALINE | $W(\text{ALINE}, \text{MCMC})$ | $3.64 \pm 0.16$ |
| ALINE | $W(\text{ALINE}, \text{GT})$ | $3.73 \pm 0.16$ |
| Reference | $W(\text{MCMC}, \text{GT})$ | $0.06 \pm 0.03$ |

The JADAI posterior is close to both the MCMC reference posterior and the ground-truth parameters, with distances on the same scale as the MCMC-to-GT reference distance. By contrast, the GMM-based ALINE posterior is substantially farther from both MCMC and GT in this two-source setting. This suggests that the diffusion-based JADAI posterior better captures the multimodal posterior structure induced by exchangeable source locations. These results complement the sPCE results in Table 1 by directly assessing the learned posterior rather than only the quality of the design policy.

# C. Experimental Details

## C.1. Metrics

### C.1.1. SEQUENTIAL PRIOR CONTRASTIVE ESTIMATION

For evaluating the performance of the policy networks exclusively (i.e., without considering downstream posterior inference results), we use sequential prior contrastive estimation (Foster et al., 2021), which is a lower bound on the sequential expected information gain and can be computed as:

$$\text{sPCE}(\pi_\phi, L) = \mathbb{E}_{p(\boldsymbol{\theta}_0)p(h_T|\boldsymbol{\theta}, \pi_\phi)p(\boldsymbol{\theta}_{1:L})} \left[ \log \frac{p(h_T \mid \boldsymbol{\theta}_0, \pi_\phi)}{\frac{1}{L+1} \sum_{l=0}^{L} p(h_T \mid \boldsymbol{\theta}_l, \pi_\phi)} \right]. \tag{43}$$

Because the upper bound of sPCE is given as $\log(1 + L)$ and thus depends on the number of contrastive samples, we adapt the evaluation settings according to the experiment, following the practices from the literature (Ivanova et al., 2021; Foster et al., 2021; Huang et al., 2026).

### C.1.2. WASSERSTEIN DISTANCE

For a quantitative evaluation of the inferred posteriors, we further compute the Wasserstein distance for the location-finding task as

$$W(p, q) = \inf_{\gamma \in \Gamma(p,q)} \mathbb{E}_{(x,y)\sim\gamma} [d(x, y)] \tag{44}$$

where $\Gamma(p, q)$ denotes the set of all couplings between two probability measures $p$ and $q$. We use solvers provided by the Python Optimal Transport (POT) library to compute this distance (Flamary et al., 2021; 2024) with a squared Euclidean metric $d$, while explicitly accounting for mode-switching in $\Gamma$.

### C.1.3. POSTERIOR QUALITY FOR IMAGE DISCOVERY

We use the structural similarity index measure (SSIM) as a sample-based quality metric for the high-dimensional posterior used in MNIST image discovery (see Section 5.3). For two images $\boldsymbol{\theta}$ and $\boldsymbol{\theta}_0$, the SSIM $\in [0, 1]$ is given by

$$\mathrm{SSIM}(i, j) = (l(i, j))^\alpha \, (c(i, j))^\beta \, (s(i, j))^\gamma, \tag{45}$$

with $i, j$ for pixel-indices, and $l$ the luminance, $c$ the contrast, and $s$ the structure of the images:

$$l(i, j) = \frac{2\mu_{\boldsymbol{\theta}}\mu_{\boldsymbol{\theta}_0} + C_1}{\mu_{\boldsymbol{\theta}}^2 + \mu_{\boldsymbol{\theta}_0}^2 + C_1} \tag{46}$$

$$c(i, j) = \frac{2\sigma_{\boldsymbol{\theta}}\sigma_{\boldsymbol{\theta}_0} + C_2}{\sigma_{\boldsymbol{\theta}}^2 + \sigma_{\boldsymbol{\theta}_0}^2 + C_2} \tag{47}$$

$$s(i, j) = \frac{\sigma_{\boldsymbol{\theta}\boldsymbol{\theta}_0} + C_3}{\sigma_{\boldsymbol{\theta}}\sigma_{\boldsymbol{\theta}_0} + C_3} \tag{48}$$

where the constants $C_1, C_2, C_3$ are computed from the images' pixel dynamic range. We use the `torchmetrics` module to compute the SSIM and leave all hyperparameters at default.

The SSIM serves as a measure of the sample reconstruction quality, which has been shown to perform better than MSE in terms of perceived accuracy (Søgaard et al., 2016; Varga, 2019; Wang & Li, 2011; Gore & Gupta, 2015). For our experiments, we compute the SSIM between posterior samples $\boldsymbol{\theta} \sim p(\boldsymbol{\theta}|\mathbf{h}_t)$ and the ground-truth MNIST images $\boldsymbol{\theta}_0$, averaging both over many samples $\boldsymbol{\theta}$ for each $\boldsymbol{\theta}_0$, as well as over all $\boldsymbol{\theta}_0$ in the validation set.

In a similar fashion, we also use `torchmetrics` to compute the normalized root mean squared error (NRMSE) with mean-normalization:

$$\mathrm{NRMSE}(\boldsymbol{\theta}, \boldsymbol{\theta}_0) = \frac{\mathrm{RMSE}(\boldsymbol{\theta}, \boldsymbol{\theta}_0)}{\mathrm{mean}(\boldsymbol{\theta}_0)} \tag{49}$$

Note that this version of NRMSE has no upper bound. For the image discovery experiment, we compute the NRMSE between posterior samples $\boldsymbol{\theta} \sim p(\boldsymbol{\theta}|\mathbf{h}_t)$ and the ground-truth MNIST images $\boldsymbol{\theta}_0$, averaging over multiple samples and all samples in the validation set. This gives a simpler, more intuitive metric for image similarity than the SSIM.

### C.2. Location Finding

A standard benchmark problem is location finding, initially proposed as an acoustic energy attenuation model (Sheng & Hu, 2005), and later adapted to the experimental design regime by Foster et al. (2021). Here we use common settings described in DAD (Foster et al., 2021), iDAD (Ivanova et al., 2021), and ALINE (Huang et al., 2026) to make fair comparisons. For self-consistency, we now describe the data generation process. The goal of the location finding experiment is to detect $K$ hidden sources $\boldsymbol{\theta} = \{\boldsymbol{\theta}_k\}_{k=1}^K$ from $T$ independent sensor measurements. At each measurement step $t$, a location $\boldsymbol{\xi}$ for the sensor placement has to be chosen. The total noise-free signal is the superposition of all $K$ signals at that location:

$$\mu(\boldsymbol{\theta}, \boldsymbol{\xi}) = b + \sum_{k=1}^K \frac{\alpha_k}{m + \|\boldsymbol{\theta}_k - \boldsymbol{\xi}\|^2}, \tag{50}$$

which is used with additive Gaussian noise $\epsilon \sim \mathcal{N}(0, 1)$ to produce the observation:

$$x = \log \mu(\boldsymbol{\theta}, \boldsymbol{\xi}) + \sigma \cdot \epsilon. \tag{51}$$

*Table 5.* Location finding experimental settings. We evaluate our method on varying prior $p(\boldsymbol{\theta})$, number of sources $K$, maximum rollout length during training $T$, number of contrastive samples $L$, and number of different ground truth samples $L_0$ based on commonly chosen settings from the literature for the location finding experiment. Results are shown in Table 1.

| Setting | $p(\boldsymbol{\theta})$ | $K$ | $T$ | $L$ | $L_0$ |
|---|---|---|---|---|---|
| iDAD (Ivanova et al., 2021) | $\mathcal{N}(0,1)$ | 2 | 10 | $5 \cdot 10^5$ | 4096 |
| | $\mathcal{N}(0,1)$ | 2 | 20 | $5 \cdot 10^5$ | 4096 |
| DAD (Foster et al., 2021) | $\mathcal{N}(0,1)$ | 2 | 30 | $1 \cdot 10^6$ | 2000 |
| ALINE (Huang et al., 2026) | $\mathcal{U}(0,1)$ | 1 | 30 | $1 \cdot 10^6$ | 2000 |

The individual source strength $\alpha_k = 1$, the background level $b = 0.1$, the maximum signal intensity $m = 10^{-4}$, and the noise level $\sigma = 0.5$ are constant throughout all settings. Similarly, we set the dimensionality of the design and parameter space to $D = 2$ resulting in four dimensional parameter $\boldsymbol{\theta} = \{[\theta_{11}, \theta_{12}], [\theta_{21}, \theta_{22}]\}$ in the case of two sources $K = 2$ and two dimensional designs $\boldsymbol{\xi} = [\xi_1, \xi_2]$. Huang et al. (2026) varied from the standard benchmark setting (Foster et al., 2021; Ivanova et al., 2021; Blau et al., 2022) by choosing a single source $K = 1$, leading to a unimodal posterior distribution which can be approximated using a marginal factorization of the posterior with Gaussians. An overview of the different settings is provided in Table 5, and additional rollout samples for a terminal horizon at $T = 30$ are shown in Figure 5.

We trained online with 200,000 simulated instances per epoch, a batch size of 256, and the Adam (Kingma & Ba, 2015) optimizer with default hyperparameters. We used 50 epochs of pretraining with a random policy ($\rho_n = 1$), followed by 400 epochs of joint training ($\rho_n = 0$). During pretraining, the learning rate was linearly ramped from $10^{-8}$ to $10^{-3}$ over the first four epochs, followed by cosine annealing down to $5 \cdot 10^{-4}$. For joint training, we linearly ramped the learning rate from $5 \cdot 10^{-5}$ to $10^{-4}$ over the first eight epochs, then cosine annealed it to $5 \cdot 10^{-5}$ for the remainder of training. All experiments were run on an NVIDIA GeForce RTX 4090, with training taking approximately 5.08 hours. Deployment for a full rollout with horizon $T = 30$ and 10,000 posterior samples took approximately 0.578 seconds. Rollout only was completed after 0.04 seconds.

## C.3. Constant Elasticity of Substitutions

A more challenging experiment from an optimal design perspective than location finding is the constant elasticity of substitutions (CES) (Arrow et al., 1961; Foster et al., 2019). The goal is to estimate a five-dimensional subjectivity parameter $\boldsymbol{\theta} = [\rho, \alpha, \log u]$ an agent has with respect to $K = 3$ different goods, where $\alpha = [\alpha_k]_{k=1}^K$ depends on the number of goods. The priors on each of the parameter components are defined as:

$$\rho \sim \text{Beta}(1, 1) \tag{52}$$

$$\alpha \sim \text{Dirichlet}(1_K) \tag{53}$$

$$\log u \sim \mathcal{N}(1, 3^2). \tag{54}$$

To estimate these values, the agent is presented with two baskets $D = 2$ filled with different numbers of each of the goods $\boldsymbol{\xi} = [\xi_1, \xi_2]$ with $\xi_1, \xi_2 \in [0, 100]^K$. The subjective utility each basket $\xi_d$ holds is defined as

$$U(\xi_d, \rho, \alpha) = \left( \sum_{k=1}^K \xi_{d,k}^\rho \alpha_k \right)^{\frac{1}{\rho}}. \tag{55}$$

The observation that can be made is proportional to the difference in subjective utilities with additive Gaussian noise $\epsilon \sim \mathcal{N}(0, 1)$ and a noise scaling term that depends on the difference of the baskets:

$$\mu(\boldsymbol{\theta}, \boldsymbol{\xi}) = u \cdot (U(\xi_1, \rho, \alpha) - U(\xi_2, \rho, \alpha)) \tag{56}$$

$$\sigma(\boldsymbol{\theta}, \boldsymbol{\xi}) = u \cdot \tau \cdot (1 + \|\xi_1 - \xi_2\|) \tag{57}$$

$$\eta = \mu + \sigma \cdot \epsilon \tag{58}$$

$$x = \text{clip}\left(\text{sigmoid}(\eta), \delta, 1 - \delta\right), \tag{59}$$

where $\tau = 0.005$ and $\delta = 2^{-22}$ (Blau et al., 2022; Huang et al., 2026). As pointed out by Foster et al. (2019), the challenge of this experiment lies in finding the sweet spot of informative designs $\boldsymbol{\xi}$. High differences between $\xi_1$ and $\xi_2$ increase

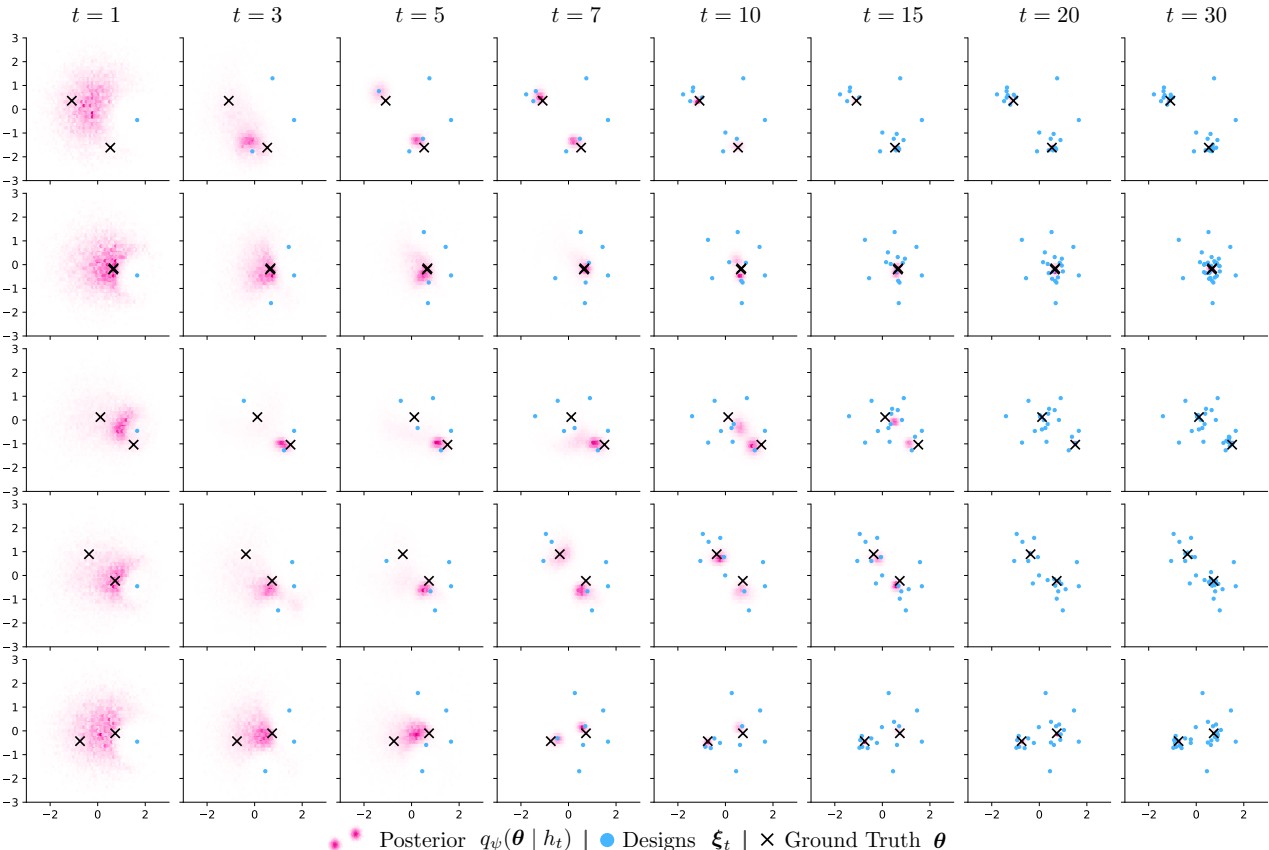

*Figure 5.* Additional rollout examples for location finding (DAD setting, cf. Table 5). Each row shows one simulated instance, and columns correspond to $t \in \{1, 3, 5, 7, 10, 15, 20, 30\}$. The Gaussian prior over source locations biases early posteriors toward the domain center. By around $t = 10$, posterior mass is typically concentrated near the true locations. Even when a location is identified early, the policy continues to explore rather than collapsing onto a single point; later measurements refine the inferred posteriors, with designs concentrating around the two source locations instead of being scattered across the domain.

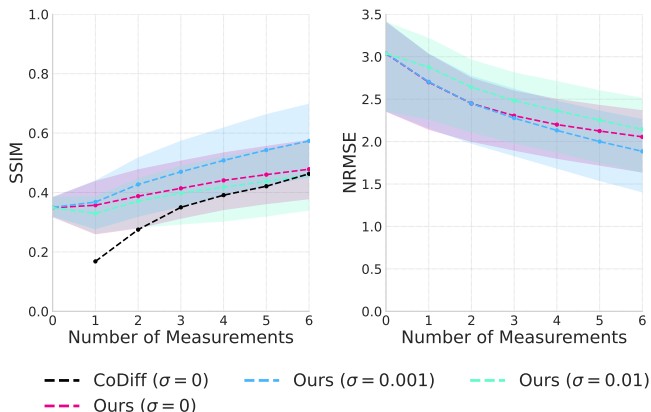

*Figure 6.* Random policy validation SSIM (↑) and NRMSE (↓) as a function of the number of measurements. Shaded bands indicate the interquartile range over the validation set. Using the random policy, our posterior samples do not differ significantly from CoDiff's in SSIM, indicating that the our method's learned-policy performance gap over CoDiff is primarily due to an improved policy rather than differences in the sampling process or network size or architectures.

the noise level $\sigma(\boldsymbol{\theta}, \boldsymbol{\xi})$ and observations potentially land on the tails of $\mathrm{sigmoid}(\cdot)$. Minor differences between the baskets might lead to similar utilities, simulating the agent's notion of indifference. Optimal designs lead to utility differences that are as far away from 0 without producing observations that land in the tail regions of $\mathrm{sigmoid}(\cdot)$.

We used the same training settings and hardware as in the location finding experiment (see above, Section C.2), but omitted pretraining and performed only joint training for 400 epochs with fully adaptive rollouts ($\rho_n = 0$). Training took approximately 2.78 hours, and deployment for a full rollout with horizon $T = 10$ and 10,000 posterior samples took approximately 0.67 seconds. Rollout only was completed after approximately 0.02 seconds.

### C.4. Image Discovery

To extend the previous experiments to higher dimensions, we also consider image discovery as proposed by Iollo et al. (2025). Here, the goal is to reconstruct an image from partial information, with each measurement unveiling only part of the image. One can imagine this experiment like standing in a dark room and trying to observe a large poster on the wall by shining light on its different parts bit by bit.

More formally, consider an unknown ground-truth image $\boldsymbol{\theta} \in \mathbb{R}^{C \times H \times W}$ with $C$ channels, as well as height $H$ and width $W$. At each experimental step, choose a location $\boldsymbol{\xi} \in [0, H] \times [0, W]$ which represents the continuous-space center of the measurement. The noise-free signal $\mu$ is then given by a smooth analog of a simple masking operation, which Iollo et al. (2025) choose as a convolution with a Gaussian kernel $G_s$:

$$\mu_{\boldsymbol{\xi},s}(x_1, x_2) = (\mathbf{A}_{\boldsymbol{\xi}}(\boldsymbol{\theta}) * G_s)(x_1, x_2) \tag{60}$$

with $\mathbf{A}_{\boldsymbol{\xi}}(\boldsymbol{\theta})$ as the masked image, smoothness parameter $s$ and $(x_1, x_2)$ the pixel locations. Iollo et al. (2025) further propose replacing the Gaussian kernel with a bivariate logistic distribution, which then simplifies the signal to

$$\mu_{\boldsymbol{\xi},s}(x_1, x_2) = [S(x_1 - \xi_1 + h; s_1) + S(\xi_1 + h - x_1; s_1) - 1] \, [S(x_2 - \xi_2 + h; s_2) + S(\xi_2 + h - x_2; s_2) - 1] \tag{61}$$

with $h$ the mask size and $S(x, s) = \frac{1}{1 + \exp(-x/s)}$ the sigmoid function with scale parameter $s$. The full measurement $\mathbf{x}$ is noisy, such that the observed value at each pixel becomes

$$x_{\boldsymbol{\xi},s}(x_1, x_2) = \mu_{\boldsymbol{\xi},s}(x_1, x_2) + \eta(x_1, x_2) \tag{62}$$

where we choose a uniform noise term $\eta \sim \mathcal{U}(0, \sigma)$. We further clamp $\mathbf{x}$ to $[0, 1]$ in order to preserve its range of values even in the presence of noise at signal values near 1. In practice, the scale $s$ and noise level $\sigma$ are small, such that the signal

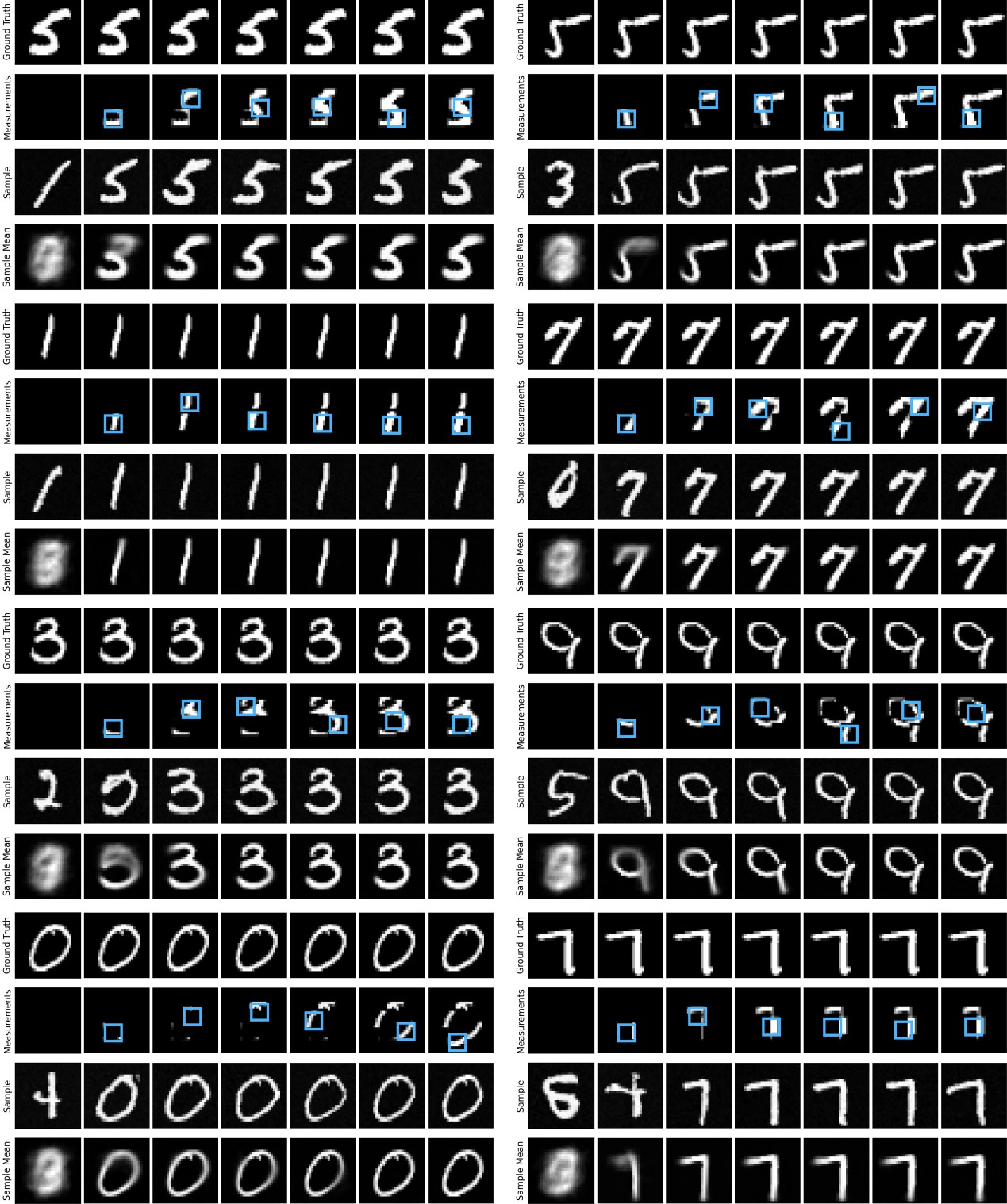

*Figure 7.* Additional validation samples for the MNIST image discovery experiment with policy, using a diffusion model with $\sigma = 0.001$. Each block shows one rollout process, where the columns represent the measurement steps. The first row shows the ground truth image. The second row shows the cumulative measurements so far, along with the newly chosen design for the current step in blue. The third row shows one posterior sample. The posterior mean over 100 samples is shown in the bottom row.

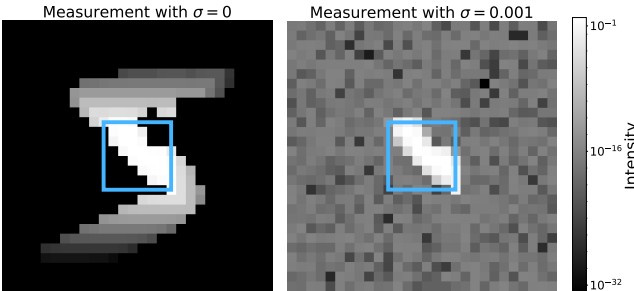

*Figure 8.* Difference in log-scale signal support for noise-free and noisy measurements in image discovery. *Left*: The noise-free measurement clearly shows the full digit shape in just one measurement, even outside of the intended measurement mask. *Right*: Using additive noise prevents useful signal outside of the mask.

dominates within, and the noise term dominates outside of the masking area. This means useful information can only be extracted from within the masking area.

We run multiple experiments with varying $\sigma$ using $s = 0.1$ and a mask size of $h = 7$ for an image size of $H = W = 28$. Just like (Iollo et al., 2025), we first run a noise-free experiment with $\sigma = 0$. However, given that the signal has support over the full image, we expect the full ground truth image to be recoverable from very small signal values in a single measurement. Therefore, we run additional experiments with $\sigma = 10^{-3}$ and $\sigma = 10^{-2}$ which destroy signal outside of the masking area. The difference in signal support is highlighted on a log scale in Figure 8.

We train for a total of 500 epochs, with the maximum number of measurements $T$, linearly scheduled from $T_i = 2$ to $T_f = 6$ within the first 5% of training steps. Similarly, we schedule the probability of design exploration $\rho_n$ with a cosine decay schedule, starting at 100% and decaying to 0% in the first 30% of training steps.

We use the AdamW optimizer (Loshchilov & Hutter, 2019) without weight decay. The learning rate is scheduled according to a `OneCycleLR` (Smith & Topin, 2019), with a maximum learning rate of $10^{-4}$, an initial division factor of 10, and a final division factor of $10^4$. The learning rate is ramped up to the maximum over the first 5% of training steps.

We use a batch size of 48, and make use of automatic mixed precision with PyTorch Lightning's precision option `"16-mixed"` (Falcon & The PyTorch Lightning team, 2019). We further clip the gradient norm at a maximum value of 5.0.

The full model is very small at only 417K parameters. Training took approximately 15.7 hours, while inference for a batch of 300 posterior samples typically takes under 3 seconds. The rollout process for all 6 measurements in a batch of one requires around 22ms. JIT-compilation further improves this to just 110µs, which is significantly faster than comparable methods on this hardware. Using larger batches could lead to additional improvements in the per-sample figure.

Apart from the results in the main section Section 5.3, we present results with the posterior approximator and a random policy in Figure 6 and additional rollout examples in Figure 7.

We expect the method to scale well to high-dimensional problems, provided sufficient hardware resources and careful handling of potentially vanishing gradients during training for very large policy networks and long rollouts (e.g., by letting the network learn the design residual instead).

## D. Neural Architectures

*Table 6.* Overview of network architectures used for policy, history, and approximate posterior in each experiment.

| Experiment | Policy | History | Inference |
|---|---|---|---|
| LF | MLP | Transformer encoder | MLP |
| CES | MLP | Transformer encoder | MLP |
| ID | ResNet + MLP | FiLM + UNet | UNet |

### D.1. MLP-based policy and posterior networks

We use a generic multilayer perceptron (MLP) as a building block for the posterior subnetworks in the LF and CES experiments and as the policy networks in all three experiments. Each MLP is composed of a sequence of hidden blocks, where each block applies a linear layer followed by a pointwise nonlinearity; depending on the experiment, dropout and layer normalization may additionally be applied. All MLPs terminate in a linear output layer. For the CES and ID policy networks, this output is then passed through a sigmoid to map the design components $\xi_t$ to $[0, 1]$. Additionally, for the CES policy network, outputs are scaled to $[0, 100]$ matching the design domain.

In the LF and CES experiments, the policy network is an MLP with four hidden layers of width $128$, using GELU nonlinearities and layer normalization after each hidden block. The diffusion-based posterior subnet in both experiments uses three hidden layers of width $512$ with GELU activations.

For the image discovery (ID) experiment, the policy network first processes the history embedding with a shallow convolutional network with residual connections and downsampling (He et al., 2016). The history encoder produces a $16 \times 28 \times 28$ feature map, which is passed through two residual blocks, each followed by $2 \times 2$ max pooling with stride $2$. Each block consists of three layers with $3 \times 3$ convolution kernels and GELU activations. In each layer, two successive convolutions with stride $1$ and padding chosen to preserve spatial resolution are applied, each followed by batch normalization, and their output is added to a residual branch that projects the input to the appropriate number of channels via a $1 \times 1$ convolution. In the first block, the channels progress as $16 \to 16 \to 16 \to 8$, in the second as $8 \to 8 \to 8 \to 4$, so that starting from $28 \times 28$, the network produces a $4 \times 7 \times 7$ feature map. This feature map is flattened and fed into the MLP policy head, which maps the resulting features to the design $\boldsymbol{\xi}_t$ via two hidden layers of width $128$ with Mish activations, dropout rate $0.05$, and residual connections between hidden blocks, followed by the sigmoid output layer.

### D.2. Transformer history encoder for scalar observations

For the LF and CES experiments, the history network is a Transformer encoder (Vaswani et al., 2017) that processes the variable length sequence of design-observation pairs with an additional normalized time index $\left\{ [\boldsymbol{\xi}_k, \mathbf{x}_k, k/T] \right\}_{k=1}^{t}$ at each rollout step $t$. This sequence is padded with zeros to the fixed maximum length $T$. Each per-step input is first projected to a $64$-dimensional embedding and summed with a learned positional embedding $P \in \mathbb{R}^{T \times 64}$. The resulting sequence is passed through a stack of $4$ Transformer encoder layers with hidden width $64$, two attention heads, and feedforward layers of width $4 \cdot 64$. A padding mask is used to prevent attention to padded positions beyond the actual sequence length. A final linear layer maps the encoder outputs to the channel dimensions of the summary states $\mathbb{R}^{T \times 64}$. The summary $\mathbf{h}_t$ is the encoder output at the current time index $t$ and is used as input to the posterior and policy networks.

### D.3. U-Net-based spatial history and inference networks for image discovery

In the image discovery (ID) experiment, both the history network and the inference network operate directly on $28 \times 28$ images and share the same Simple U-Net backbone architecture, applied to different inputs. The history path first uses a FiLM-based encoder to build per–step spatial features from design–observation pairs $(\boldsymbol{\xi}_t, \mathbf{x}_t)$, whereas the inference path conditions the U-Net on the spatial summary state $\mathbf{h}_t$ and an additional diffusion-time embedding.

At step $t$, the design $\boldsymbol{\xi}_t \in [0, 1]^2$ and the corresponding noisy observation $\mathbf{x}_t \in \mathbb{R}^{1 \times 28 \times 28}$ are passed through a FiLM network (Perez et al., 2018). Rather than concatenating $\boldsymbol{\xi}_t$ as additional channels, this network uses FiLM to modulate convolutional feature maps from the observations by the corresponding 2D design inputs. Concretely, the FiLM encoder consists of 3 convolutional FiLM blocks. In each block, the input is processed by 2 successive $3 \times 3$ convolutions with stride $1$ and padding chosen to preserve spatial resolution, both using $32$ intermediate channels and ReLU activations. The resulting features are batch-normalized and FiLM-modulated, and a $1 \times 1$ residual projection maps the block input to the output channels (32 in the first two blocks and 4 in the last) before the residual is added. FiLM modulation parameters $(\beta, \gamma)$ are produced from $\boldsymbol{\xi}_t$ by an MLP with two hidden layers of width $128$, and applied to the feature maps as $\gamma \odot \mathbf{z} + \beta$, followed by a final ReLU and addition of the residual projection. Starting from $\mathbf{x}_t \in \mathbb{R}^{1 \times 28 \times 28}$, this yields a per-step feature map $\mathbf{z}_t \in \mathbb{R}^{4 \times 28 \times 28}$.

Given the sequence of FiLM output features $(\mathbf{z}_1, \dots, \mathbf{z}_t)$, information is aggregated over time by summing the features, $\mathbf{z}_{1:t} = \sum_{k=1}^{t} \mathbf{z}_k$, and feeding $\mathbf{z}_{1:t}$ into a two-stage convolutional U-Net. This U-Net has 8- and 16-channel stages and uses 4 and 2 residual blocks per stage, respectively. Each residual block follows the Simple Diffusion U-Net design (Hoogeboom et al., 2023), applying layer normalization, a SiLU activation, a $3 \times 3$ convolution, a second layer normalization, SiLU, and

a final $3 \times 3$ convolution, with a residual connection from the block input. We only use self-attention in the middle block. The down path alternates between residual blocks and $2 \times 2$ spatial downsampling, collecting skip features, while the up path mirrors this structure with $2 \times 2$ upsampling and additional residual blocks that fuse the corresponding skips. A final group normalization, SiLU activation, and $3 \times 3$ convolution map the output to 16 channels, yielding the spatial summary state $\mathbf{h}_t \in \mathbb{R}^{16 \times 28 \times 28}$.

For posterior inference over the underlying digit, we reuse the same Simple U-Net architecture with three modifications compared to the history encoder. First, the stage channel widths are increased from $(8, 16)$ to $(16, 32)$. Second, the input now concatenates the ground truth digit $\boldsymbol{\theta} \in \mathbb{R}^{1 \times 28 \times 28}$, the current summary state $\mathbf{h}_t \in \mathbb{R}^{16 \times 28 \times 28}$, and a scalar diffusion time (log-SNR) embedding. The time input is mapped through a $4$-dimensional sinusoidal embedding followed by a two-layer MLP, and the resulting scalar feature is broadcast to a $1 \times 28 \times 28$ map. Concatenating $\boldsymbol{\theta}$, $\mathbf{h}_t$, and this time channel yields an 18-channel input on which the U-Net operates, with the same residual blocks, skip connections, and single bottleneck self-attention as in the history network. Finally, the last projection maps back to a single channel, so that the output again lies in $\mathbb{R}^{1 \times 28 \times 28}$. In the diffusion-based ID variant, this conditional U-Net parameterizes a $\mathbf{v}$-prediction model with a cosine noise schedule, whereas in the flow-matching variant, the same backbone is used to parameterize as a conditional vector field $v$. The corresponding objectives and integration schemes are described in Section A.2 and Section A.3, respectively.

## E. Code availability

The source code is publicly available under https://github.com/bayesflow-org/JADAI.

