# OpenReview forum: "JADAI: Jointly Amortizing Adaptive Design and Bayesian Inference"
_ICML.cc/2026/Conference — ICML 2026 regular_

### Official Review · Reviewer_wH8J · 2026-03-03

**Soundness:** 2
**Presentation:** 2
**Significance:** 3
**Originality:** 2
**Overall Recommendation:** 4
**Confidence:** 3

**Summary:**

This paper introduces a method for amortized adaptive design and Bayesian inference, framed as an end-to-end training process that maximizes information gain. The approach integrates three key components:
- A policy network (for design proposal),
- A history network (to compress past designs and observations),
- An inference network.

Notably, the authors employ **expressive diffusion models** as inference networks, diverging from traditional normalizing flows.

The method is evaluated across three benchmarks auch as **Location Finding**, **Constant Elasticity of Substitution**, and **MNIST Image Discovery**, where it is compared to prior works using metrics such as sequential prior contrastive estimation, **Structural Similarity Index Measure (SSIM)**, and **normalized root-mean-square error (NRMSE)**. The experiments demonstrate strong performance, often surpassing existing approaches.

**Compliance With Llm Reviewing Policy:**

Affirmed.

**Final Justification:**

While I am not an expert in this field, the paper appears both novel and technically sound, with strong empirical performance demonstrated across the evaluated benchmarks. The problem description is clearly and effectively presented; however, the methodology section could benefit from further refinement in its writing.
Initially, I had some concerns regarding the Barber–Ågakov variational lower bound and some other equations, but these were satisfactorily addressed during the rebuttal. As a result, I have increased my score from 3 to 4.

**Key Questions For Authors:**

1. **Equation 15 vs. Equation 12:**
   Why is **log p** used in Equation 15, while **log q** is used in Equation 12 for the inference network? Is there a specific reason for this discrepancy?

2. **Potential Error in Equation 13:**
   The prior **p(θ)** appears on the left side of Equation 13 but is absent on the right. Thus, I do not see that this identity is correct.

3. **Theoretical Validity:**
   Does the **Barber-Ågakov variational lower bound** remain valid when a diffusion model is used as the inference network?

4. **Role of Diffusion Models:**
   Could the authors elaborate on how the diffusion model integrates into the framework?

5. **Stopped Gradients**
Why are gradients partially stopped in Equation 18, and what are the implications for training dynamics?

5. **History Network and Summary Statistics:**
   While truncating backpropagation through time (BBTT) for memory efficiency is understandable, the paper states that the history state is detached after each step (Section 3.4). How can the history network learn meaningful summary statistics under these conditions?


## Suggestions for Improvement
- Provide a more detailed explanation of diffusion model’s role in the appendix, including its impact on the variational lower bound.
- Expand the discussion of implementation choices into the appendix (e.g., gradient stopping, rollout scheduling) to justify their necessity and effects.
- Address potential ambiguities in equations (e.g., Equations 12, 13, and 15) to ensure clarity for readers.

## Final Thoughts
The paper presents a promising approach with strong empirical results, but its theoretical and methodological clarity could be enhanced.

**Strengths And Weaknesses:**

## Strengths
1. **Clear Background and Motivation:** The paper provides a thorough and accessible introduction to the problem, contextualizing the proposed method within the broader literature.
2. **Empirical Performance:** The method exhibits robust performance across benchmarks, frequently outperforming prior work. The experimental setup and comparisons are well-justified.

---

## Weaknesses and Concerns
1. **Lack of Clarity on Diffusion Models:**
   - The paper insufficiently explains the role and integration of diffusion models within the inference network. Their usage is not well-motivated, and the theoretical implications, particularly regarding the validity of the **Barber-Ågakov variational lower bound**, remain unclear.
   - If the bound no longer holds, the method’s theoretical foundation may be compromised, potentially leading to instability.

2. **Algorithmic and Implementation Details:**
   - **Algorithm 1** does not adequately clarify the diffusion model’s role, leaving critical steps ambiguous.
   - The rationale for stopping gradients in **Equation 18** is unclear and should be explained in more detail in the appendix.

3. **Complexity and Design Choices:**
   - The method is highly intricate, incorporating numerous tweaks (e.g., stopping gradients in the telescoping sum, stopping gradients in backpropagation through time, rollout length scheduling, and exploration via design prior mix-in).

---

> ### Author Rebuttal · Authors · 2026-03-31
>
> # Clarifying the integration of Diffusion Models
> We understood the reviewer's point to be a lack of clarity regarding how the diffusion model is integrated into JADAI (Weakness 1 and Q4) and where it appears in Algorithm 1 (Weakness 2). We will address this below. If we are mistaken and the Reviewer is asking more generally about the motivation for using implicit density approximators, please refer to the answer for Reviewer Hf83.
>
> Instead of optimizing the KL divergence, which appears in the general form of the BA directly through explicit density approximation methods like normalizing flows, we demonstrate that it is possible to use implicit density approximation objectives implied by diffusion- or flow-matching-based learning algorithm. However, regardless of the objective induced by the model family, its loss evaluations are used to calculate gradients for updating all three networks. This is why, in algorithm 1, on lines 3 and 7, we write $\mathcal{L}\_{post}$ rather than specifying the exact loss objective being evaluated (eg. $\mathcal{L}\_{post}^{diff}$ or $\mathcal{L}\_{post}^{flow}$), since this varies across the model families we consider.
> For example, for the LF and CES experiments, we have used a diffusion loss objective (eq. 27), and for MNIST, a flow matching loss objective (eq. 34), both plugged into the placeholder $\mathcal{L}_{post}$.
>
> ---
>
> # Relation to BA
> We thank the reviewer for raising this important question regarding the relation to the BA bound (Weakness 1, Q3) and would like to refer to the answer to Reviewer WQVK.
>
> ---
>
> # Detaching gradients and meaningful summary statistics
>
> The reviewer pointed out a lack of clarity about how detaching of gradients influences the training dynamics (Weakness 2, 3 and Q5) and how the history network is still enabled learning meaningful summary statistics (Q6). We agree that the discussion on these design choices should have been more thorough and will update the submission accordingly.
>
> The history is detached at each step to prevent nested backpropagation through time (BPTT), as discussed in Section 3.4; otherwise, it would lead to BPTT at every rollout step (i.e., $T$ BPTTs) and unstable training (Appendix B).
> By detaching the history state $h_{t-1}$ before the policy, and the individual posterior loss terms reused in the next rollout step (eq. 16), the $T$ gradients from the posterior-loss evaluations along the rollout are computed independently (eq. 18) (see discussion on relation to BA). This update is still informative because the history encoder uses prior and current tokens to produce the current history state, which is then used to evaluate the stepwise posterior loss. Consequently, the history network is informed of the quality of each encoding at each rollout step, as in regular summary-to-inference pipelines [1, 2].
>
> [1] Radev et al. (2020), “BayesFlow: Learning complex stochastic models with invertible neural networks”
>
> [2] Chen et al. (2023), “Is learning summary statistics necessary for likelihood-free inference?”
>
> ---
>
> # Design Choices
> Thank you for raising these points on clarifying the design choices. We will state this more clearly in the main part using the extra page in the camera-ready version. Regarding the choice of design prior mix-in during training, we would like to refer to the answer we provided to Reviewer MLg2.
>
> The rollout-length curriculum serves a different role. It is only used very early in training, for roughly 0.2% of total update steps for efficiency reasons. When the networks are still near initialization, running the full rollout already is often unnecessary because later rollout-step losses are not yet very informative.
>
> Sampling the rollout length reweights training toward short rollout regimes that contain less observations, and are harder for the inference task. The motivation for sampling the rollout lengths is thus similar to diffusion schedules that emphasize noisier regimes more strongly. In Appendix B we discussed this tradeoff. Always training at the maximum horizon improves terminal performance, but degrades shorter-horizon behavior.
>
> ---
>
> # Eqs. 12 vs. 15, 13
> We thank the reviewer for the clarification request (Q1) and for pointing out the discrepancy in eq.13 (Q2).
>
> Eq. 12 is the BA variational bound and therefore must be written in terms of the approximate posterior $q_\psi(\theta \mid h_T)$. By contrast, eq. 15 was intended to illustrate the score-based analogue arising when moving from explicit densities to implicit models, with its neural approximation introduced later in the diffusion appendix.
>
> The correct decomposition in eq. 13 is
>
> $$ \log \frac{q_\psi(\theta \mid h_T)}{p(\theta)} = \sum_{t=1}^{T} \log \frac{q_\psi(\theta \mid h_t)}{q_\psi(\theta \mid h_{t-1})} + \log \frac{q_\psi(\theta \mid h_0)}{p(\theta)}. $$
>
> We will correct Eq. (13) accordingly.

---

> > ### Author Rebuttal · Reviewer_wH8J · 2026-04-03
> >
> > I thank the authors for their answers.
> > I have difficulties following the answer to Reviewer WQVK regarding the BA bound.
> > For example, "$sCEE_t(\Phi)$" seems not to be defined in the rebuttal nor in the paper, so it is hard for me to follow the proof.
> > Could the authors please give a more detailed proof below?

---

> > > ### Author Response · Authors · 2026-04-03
> > >
> > > We are pleased to provide the more detailed proof below and hope it addresses the reviewer's remaining question about the relationship between JADA's objective and the BA bound. As the reviewer pointed out, the sCEE from [2] was not mentioned explicitly in the paper (and we will change that), but it is essentially the log-density ratio in eq. 12, where the true posterior density $p$ from the TEIG (eq. 9) is replaced with the approximate density $q\_\psi$.
> > >
> > > ## BA per horizon
> > >
> > > [2] introduced the sequential cross-entropy estimator (sCEE) (eq. 12) as the posterior-form TEIG bound (eq. 9; and see [1]). The same construction applies at any $t \leq T$. The TEIG at rollout step $t$ is
> > >
> > > $$\mathrm{TEIG}\_t := \mathbb{E}_{p(\theta, h_t)}\bigg[\log \frac{p(\theta \mid h_t)}{p(\theta)}\bigg], \tag{D.1}$$
> > >
> > > Following [2] (Theorem 1, Corollary 2), replacing $p(\theta \mid h_t)$ by any approximate posterior $q\_\psi(\theta \mid h_t)$ yields
> > >
> > > $$\mathrm{TEIG}\_t = \underbrace{\mathbb{E}\_{p(\theta, h\_t)}\bigg[\log \frac{q\_\psi(\theta \mid h\_t)}{p(\theta)}\bigg]}\_{\mathrm{sCEE}\_t(\psi)}+ \delta\_t(\psi), \qquad \delta\_t(\psi) := \mathbb{E}\_{p(h_t)}\big[\mathrm{KL}\big(p(\theta \mid h_t) \Vert q\_\psi(\theta \mid h\_t)\big)\big] \ge 0, \tag{D.2}$$
> > >
> > > so that $\mathrm{sCEE}\_t(\psi) \le \mathrm{TEIG}\_t$ with equality iff $q\_\psi = p(\theta \mid h\_t)$.
> > > This inequality is agnostic to how $q\_\psi$ was obtained; it depends only on the resulting posterior approximation.
> > >
> > > The gap $\delta_t(\psi)$ comprises the standard approximation error of a finite-capacity neural network, present for any posterior model including normalizing flows, as well as an additional component when $q_\psi$ is trained via score estimation ($\mathcal{L}\_{\mathrm{post}}^{\mathrm{diff}}$, eq. 27) or flow matching ($\mathcal{L}\_{\mathrm{post}}^{\mathrm{fm}}$, eq. 34) rather than direct density optimization [3, 4, 5]. Both components can be reduced over the course of training and, in all cases, $\mathrm{sCEE}\_t(\psi) \le \mathrm{TEIG}\_t$ remains valid regardless of the magnitude of $\delta\_t(\psi)$.
> > >
> > > ## Summing TEIGs
> > >
> > > Summing (D.2) over $t = 0, \dots, T$:
> > >
> > > $$\sum\_{t=0}^{T} \mathrm{sCEE}\_t(\psi) = \sum\_{t=0}^{T} \mathrm{TEIG}\_t - \sum\_{t=0}^{T} \delta\_t(\psi) \le \sum\_{t=0}^{T} \mathrm{TEIG}\_t. \tag{D.3}$$
> > >
> > > Since $\mathrm{TEIG}\_t$ does not depend on $\psi$, maximizing $\sum\_t \mathrm{sCEE}\_t(\psi)$ with respect to $\psi$ is equivalent to minimizing $\sum\_t \delta\_t(\psi)$. The detached telescoping provides exactly this multi-horizon optimization.
> > > To see this, we can define an alternative objective
> > >
> > > $$\mathcal{L}^{\mathrm{sum}}(\psi) := \sum\_{t=0}^{T} \mathbb{E}\_{p(\theta, h\_t)}\big[\ell\_t \big], \tag{D.4}$$
> > >
> > > where $\ell\_t = \mathcal{L}\_{\mathrm{post}}(\theta, h\_t)$ is the per-step posterior loss (eq. 11). Recall (eq. 16) that the detached utility $u\_T = \sum\_{t=0}^{T}(\ell\_{t-1}^* - \ell_t)$ with $\ell\_{-1} := 0$ telescopes to $u\_T = -\ell\_T$ as a scalar, but the gradient with respect to $\psi$ does not telescope due to the stop-gradient (eq. 18):
> > >
> > > $$\nabla\_\psi \mathcal{L}(\psi) = \sum\_{t=0}^{T} \nabla\_\psi \mathbb{E}\big[\ell\_t\big] = \nabla\_\psi \mathcal{L}^{\mathrm{sum}}(\psi). \tag{D.5}$$
> > >
> > > The stop-gradient thus ensures that $q\_\psi$ receives posterior-training signal from every rollout step $t = 0, \dots, T$ (not only the terminal signal) independently (i.e., no nested BPTT due to the optimization along the rollout), even though the scalar objective $\mathcal{L} = \mathbb{E}[\ell\_T]$ involves only the terminal loss.
> > >
> > > In the density case ($\ell\_t = \ell\_t^{\mathrm{flow}} = -\log q_\psi(\theta \mid h_t)$), the alternative objective $\mathcal{L}^{\mathrm{sum}}(\psi)$ reduces to $\sum\_t \mathbb{E}[-\log q\_\psi(\theta \mid h\_t)]$. Since no $\mathrm{TEIG}\_t$ depends on $\psi$, it follows from (D.2) that
> > >
> > > $$\nabla\_\psi \mathcal{L}^{\mathrm{sum}}(\psi) = \sum\_{t=0}^{T}  \nabla\_\psi  \delta\_t(\psi), \tag{D.6}$$
> > >
> > > Detaching the gradients (D.5) exactly minimizes the summed BA gaps $\sum\_t \delta\_t(\psi)$.
> > >
> > > In contrast, in the score-based case, $\ell\_t$ is instantiated as $\mathcal{L}\_{\mathrm{post}}^{\mathrm{diff}}$ (eq. 27) or $\mathcal{L}\_{\mathrm{post}}^{\mathrm{fm}}$ (eq. 34) rather than a density-based objective i.e. $-\log q\_\psi(\theta \mid h\_t)$. As noted in Section 3.2, the Barber-Agakov interpretation as an explicit log-density ratio optimization no longer holds directly, so the exact identity (D.6) does not apply. However, the gradient (D.5) remains unchanged, i.e. the detached telescoping still accumulates $\nabla\_\psi \ell\_t$ across all $t \leq T$, serving as a surrogate for minimizing the summed BA gaps $\sum\_t \delta\_t(\psi)$, where each gradient step reduces $\delta\_t(\psi)$ and tightens $\mathrm{sCEE}\_t(\psi)$ toward $\mathrm{TEIG}\_t$.
> > >
> > > ---
> > >
> > > For the citations, we would like to refer the reviewer to the answer for reviewer WQVK.

---

### Official Review · Reviewer_MLg2 · 2026-03-05

**Soundness:** 3
**Presentation:** 4
**Significance:** 3
**Originality:** 3
**Overall Recommendation:** 5
**Confidence:** 4

**Summary:**

This paper proposes a framework that jointly amortizes Bayesian adaptive design and posterior inference by training a policy network, history/summarization network, and inference network end-to-end. The method targets problems within Bayesian experimental design that rely on simulation-based inference, that is, dealing with implicit likelihoods and (possibly) complex/high-dimensional posteriors. The method uses diffusion-based posterior estimators and optimizes an incremental posterior-loss objective, rather than directly optimizing total EIG.

**Compliance With Llm Reviewing Policy:**

Affirmed.

**Final Justification:**

The changes the authors plan to make (adding practical guidance, additional ablations, posterior-quality metrics via Wasserstein distance, and some other clarifications) address all my concerns.

**Key Questions For Authors:**

My main questions concern the experimental results and their discussion:

1. **Sensitivity to $\rho_n$ (exploration / random-design mix-in).**
As I mentioned before, it seems that training behavior is sensitive to this schedule, and the preferred regime differs by benchmark (e.g., random-design pretraining helps in Location Finding, while CES training with random designs is unstable/collapsing, and Image Discovery uses an annealed mixed-policy schedule). Could the authors discuss how sensitive performance is to $\rho_n$, and provide practical guidance for choosing/scheduling it in new problems?

2. **Metrics to assess posterior inference in the LF and CES examples.** In the Image Discovery example, the paper includes an inference-quality metric (NRMSE), which helps assess how well $\theta$ is recovered. In the LF and CES examples, however, the evaluation appears to be only the sPCE. In the LF case, Figure 2 includes parameter-related visualizations, but these mainly illustrate how designs are distributed over time. The bottom-right panel shows posterior samples, but the explanation is brief and difficult to interpret quantitatively. I would appreciate an additional metric (e.g., NRMSE or similar) for LF and CES.

3. **Missing table entries / table notation clarity**. In Table 1, some competitor methods only have results for a subset of columns (e.g., iDAD only for the first two LF columns), and in Table 2 CoDiff has NRMSE marked unavailable. Could the authors clarify why these values are missing/unavailable? Also, the shorthand table headers in Table 1 (e.g., “10T 2K 5·10^5L”) are difficult to interpret in the main text without additional explanation or going to the appendix.

4. **Training cost / scalability comparison**. The appendix reports nontrivial training times (hours to ~48h depending on the benchmark), and also deployment/posterior sampling timings. It would be helpful to compare training cost more explicitly against competitors, and to discuss computational bottlenecks/scaling behavior (e.g., dependence on design dimension, parameter dimension, rollout horizon, posterior sample count).

**Limitations:**

yes

**Strengths And Weaknesses:**

**Soundness**. Overall, I find the methodology clearly presented and reasonably well justified. The objective construction and training procedure are explained in sufficient detail, and the experimental setup is generally strong. Results are mostly discussed appropriately.

That said, I think some experimental aspects would benefit from further discussion. For example, the role of the exploration / random-design mix-in schedule ($\rho_n$) appears important for successful training and may be problem-dependent. Since this seems to be a sensitive hyperparameter, I would appreciate more guidance on how it should be selected in practice when prior information about the task is limited.

**Presentation**. The paper is clearly written, easy to follow, and well-positioned with respect to related work.

**Significance**.
The paper addresses a relevant and timely problem in Bayesian experimental design: enabling fast deployment via amortization while accurately approximating complex/high-dimensional posteriors. This is especially relevant in the context of BED, where there is growing interest in implicit-likelihood (simulation-based) settings, in which both inference and adaptive design can be computationally challenging.

**Originality**.
The contribution builds on ideas of prior amortized methods (e.g., ALINE) and extends them to jointly perform adaptive design selection and posterior inference. This is a valuable extension, particularly in simulation-based settings and in problems with complex/high-dimensional posteriors.

---

> ### Author Rebuttal · Authors · 2026-03-31
>
> # Practical guidance for design choices and ablation
> We thank the reviewer for pointing out the lack of concrete discussion on design choices. Our main takeaway is that the preferred choice in the design-prior regime depends on how informative random designs are for the task.
>
> When the design space is relatively simple, i.e., random designs produce useful observations, pretraining with the design prior is effective, as we used in LF (Section 5.1) and in consistency with prior work [1].
> When random designs are somewhat informative but broad early coverage is still helpful, we have chosen an annealed mix-in as a compromise (Image discovery, Section 5.3).
> When large parts of the design space are uninformative and random-design training is not useful, we started jointly (training can even become unstable, e.g., CES, Section 5.2).
>
> In additional ablations running with the LF experiment (settings as described in Appendix B), joint training from the start and an MNIST-style mix-in performed very similarly ($12.348 \pm 0.068$) and ($12.352 \pm 0.044$) (vs. vanilla Table 3 ($12.708 \pm 0.081$)). Thus, we suggest that these regimes should be viewed as practical choices matched to the task.
>
> If only limited prior information about the task is available, we suggest that training in these different regimes should proceed in the order they are described above.
>
> We will expand Appendix B with these additional ablation results and also add a paragraph that mentions these design considerations more explicitly as well as describe choices related to the rollout-length during training, as explicitly mentioned by Reviewer wH8J.
>
> ---
>
> # Posterior quality metrics
> We thank the reviewer for this very helpful suggestion. We agree that, compared with the MNIST experiment where SSIM and NRMSE are already reported, the current LF/CES results make posterior quality harder to assess quantitatively. To complement the policy evaluation, we therefore computed Wasserstein distances (as requested by reviewer Hf83) in the LF setting used in Appendix B (Table 3), comparing learned posterior samples against both high-quality MCMC samples and the ground truth.
>
> For JADAI at $T=30$, we obtain
> $W(\text{JADAI}, \text{MCMC}) = 0.07 \pm 0.01$
>
> $W(\text{JADAI}, \text{GT}) = 0.09 \pm 0.03$
>
> $W(\text{MCMC}, \text{GT}) = 0.11 \pm 0.03$
>
> For comparison, we also evaluated ALINE [1]
> $W(\text{ALINE}, \text{MCMC}) = 3.64 \pm 0.16$
>
> $W(\text{ALINE}, \text{GT}) = 3.73 \pm 0.16$
>
> $W(\text{MCMC}, \text{GT}) = 0.06 \pm 0.03$
>
> These additional results provide a direct posterior-quality metric for LF and suggest that JADAI’s learned posterior is close to both MCMC and ground truth, whereas the GMM-based ALINE posterior is not. We will add these results to the appendix to complement the existing sPCE evaluations and make posterior quality in the low-dimensional benchmarks more explicit.
>
> ---
>
> # Clarifying baseline comparison Table 1
> We agree that Tables 1 and 2 should be clearer. In Table 1, all competitor values were taken as reported in the corresponding prior work, and missing entries indicate settings that were not reported there. We will therefore revise the table captions and headers to make the provenance of the reported values explicit, clarify the shorthand setting labels, and mark missing entries more clearly as “not reported in prior work”.
> Our goal was to compare JADAI against the benchmark settings currently available in the literature. We agree that such a benchmark would be valuable for the community, but constructing it is beyond the scope of the present submission.
>
> ---
>
> # Computational costs
> We thank the reviewer for pointing out our inconsistency of reporting training times on different machines and will update the MNIST training time to reflect a run on the same GPU as the other experiments (LF: 5.08h, CES: 2.78h), bringing it to approximately 15.7h.
>  We will also add separate rollout-only times. Rollout alone takes 0.04s on LF and 0.02s on CES, while the corresponding totals are 0.578s and 0.67s.
>
> For a more thorough overview, we would like to direct to Table 2 in ALINE [1], showing that our rollout times are competitive and training times are usually shorter.
> For CoDiff’s MNIST example (Section 5.3), the full 6-step rollout takes about 0.01s, which remains below their reported 7.2s per measurement step [2].
>
> More generally, the main bottleneck is the diffusion-model sampling procedure, which currently uses 1000 steps as a generic choice. This can be substantially reduced by using fewer steps (if the problem allows) or more sophisticated post-hoc distillation. We will update the appendix to make this runtime breakdown more explicit.
>
> ---
>
> [1] Huang et al. (2025), “ALINE: Joint Amortization for Bayesian Inference and Active Data Acquisition”
>
> [2] Iollo et al. (2025), "Bayesian Experimental Design via Contrastive Diffusions"

---

> > ### Author Rebuttal · Reviewer_MLg2 · 2026-04-01
> >
> > Thank you for the rebuttal. I think the concrete additions you plan to include will improve the manuscript. To summarise, you will add:
> >  1. practical **guidance/discussion on the design-prior regime** (which depends on the task at hand) **and on the rollout length** during training, together with **additional ablation results**;
> >  2. a metric to **assess posterior inference (Wasserstein distances)** compared to MCMC samples and ground truth;
> >  3. **clarifications in Tables 1 and 2**; and
> >  4. more detailed **information/discussion on computational cost**.
> >
> > Overall, these changes address my main concerns and reassure my positive assessment of the paper.

---

> > > ### Author Response · Authors · 2026-04-06
> > >
> > > We would like to thank the reviewer for their thoughtful engagement during the rebuttal phase. We appreciate the constructive feedback and are glad that the planned clarifications and additions addressed the main concerns.

---

### Official Review · Reviewer_Hf83 · 2026-03-13

**Soundness:** 2
**Presentation:** 2
**Significance:** 1
**Originality:** 1
**Overall Recommendation:** 2
**Confidence:** 3

**Summary:**

This paper introduces JADAI, a framework to jointly optimise adaptive design policies and inference policies for Bayesian experiment design tasks. They perform empirical work comparing their method against previous methods on some synthetic BED tasks.

**Compliance With Llm Reviewing Policy:**

Affirmed.

**Key Questions For Authors:**

Can you help me apprecaite why you think this is a significant problem, and why you think your method addresses it in totality? What is the strongest case to be made for jointly optimising inference and design? Can you provide some emprical work showing specifically the value of jointed optimisation and not just post-hoc as you criticse? e.g. can you construct some sort of ablation of your method to show the value specifically of jointed optimisation? I am open to changing my mind on this paper but I dont think this work makes a strong case for the topic the authors are interested in.

**Limitations:**

No limtations discussed

**Strengths And Weaknesses:**

Strengths:
- the authors did a good job identifying and evaluating relevant prior work

Weakness:
- the authors do not provide a compelling motivation for their entire work: they claim that most prior methods for amortized BED tasks do not also amortize an inference network, but provide no compelling need for this. For example, in the introduction, the motivation for their work is “As a result, design and inference are typically treated as separate tasks”. I did not find this to be persuasive in favor of their work. I expected the authors to argue why they ought to be treated jointly at the very least. The fact that something can be done is not itself a compelling reason to do it.

- The authors also do not evaluate the quality of the learned inference network in a systematic fashion (e.g. by calculating divergences or wasserstein distances w.r.t. high quality MCMC estimates) so the primary research contribution they seek to make can't be easily assessed.

- Their comparison is also incomplete and unclear: it's not obvious to me why they are comparing different methods with different size informaiton sets, this seems like a very unfair comparison. In table 1, why have all methods not been provided with the same information sets sizes so we can see degradation with time?

- also, why focus on diffusion? the motivation for this is also not particualrly copelling; they cite multi-modal posteriors as a justificaiton, but they don't cite a context where that is both important and sufficiently complex to warrant a diffusion policy over a GMM (e.g. their examples show multimodality that can be well captured by a GMM)

- sevearl places the language is inapproriate for an ICML paper, e.g. "This is the gist”

---

> ### Author Rebuttal · Authors · 2026-03-31
>
> # Motivation for joint optimization
> We thank the reviewer for highlighting the need for further motivating our method. In Bayesian experimental design, the objective is to choose designs that improve posterior inference about $\theta$: the EIG is defined through posterior shrinkage, and in the adaptive setting, each new design is valuable only insofar as it improves the current posterior. Joint optimization is therefore not introduced merely because it is possible, but because posterior inference is the underlying task for which Bayesian adaptive design is optimized. JADAI makes this explicit by using the per-step posterior loss as the shared training signal for the policy, history, and inference networks, while also enabling exposure of $q_\psi(\theta \mid h_t)$ after every measurement step at test time.
>
> This is especially important in the likelihood-free regime. As discussed in Section 4, traditional BAD pipelines optimize designs and then perform a separate Bayesian inference step, typically using MCMC/SMC or, when the likelihood is intractable, a separate simulation-based inference procedure. JADAI addresses this gap directly by amortizing the posterior estimator that the scientist ultimately wants to query. Accordingly, the relevant deployment times mentioned are not just rollout speed, but the total cost of rollout and posterior inference. This distinction is particularly important in high-dimensional settings such as experiment 3, where a separate inferential stage can be a substantial bottleneck. Please also see the more detailed computational cost breakdown provided in response to reviewer MLg2.
>
> Finally, the significance of JADAI is not just that design and inference are optimized jointly. JADAI’s new objective makes expressive modern SBI backbones (i.e., diffusion/flow matching) accessible in adaptive settings. We believe this is timely because modern SBI has moved away from normalizing flows and uses exactly these free-form posterior approximators for complex, high-dimensional, and multimodal inference problems [1] (please see the discussion on diffusion models below).
>
> In sum, our claim is not that every BED problem must be solved jointly. It is that, in adaptive likelihood-free settings where posterior inference is both the optimization target and a deployment bottleneck, joint amortization is a very practical solution to the problem.
>
> ---
>
> # Evaluating posterior quality
> We thank the reviewer for this helpful suggestion and performed additional computations to assess the posterior quality metrics using the suggested Wasserstein distance. Please refer to the answer for reviewer MLg2.
>
> ---
>
> # Clarifying baseline comparison Table 1
> We agree that Table 1 can currently be misread, but we do not think the intended comparisons are unfair. Please refer to the answer for reviewer MLg2.
>
> ---
>
> # Diffusion model as inference backbone
> We appreciate the reviewer's comment on questioning the use of diffusion models generally. First, we want to clarify that diffusion is used in JADAI as the posterior estimator, not as the policy. We also respectfully disagree with the claim that the paper lacks relevant context for the usefulness of such models. In the introduction and related work sections, we cite diffusion, flow matching, and consistency models, along with a recent SBI review, specifically as modern approaches for handling complex, multimodal, and high-dimensional posteriors in the SBI setting.
>
> At the same time, we do not claim that diffusion is uniquely necessary for every benchmark in this paper. Our point is that JADAI removes the explicit-density restriction of prior unified amortized design and inference methods and thereby makes expressive implicit posterior estimators available in adaptive design. This is most clearly reflected in section 5.3 (Image Discovery), the high-dimensional benchmark introduced by CoDiff [2], where the inference target is the digit image as a whole, and the posterior must capture dependencies across the image. We agree with the reviewer that studying GMM-based approximations in such settings could be interesting, but we believe a joint posterior approach with free-form generative backbones might be better suited to this regime.
>
> ---
>
> # Language
> We thank the reviewer for pointing this out and we will update the wording accordingly.
>
> ---
>
> # Key Questions
> We hope that the discussion above clarifies the reviewer’s questions regarding the significance of joint optimization, the strongest case for coupling design and inference, and the distinction between joint amortization and post-hoc inference. The comments were already helpful in revising the manuscript. If a specific part of this motivation still seems unconvincing, we would appreciate the reviewer indicating which aspect remains unresolved.
>
> ---
>
> [1] Arruda et al. (2025), “Diffusion Models in Simulation-Based Inference: A Tutorial Review”
>
> [2] Iollo et al. (2025), “Bayesian Experimental Design Via Contrastive Diffusions”

---

> > ### Author Rebuttal · Reviewer_Hf83 · 2026-04-06
> >
> > I appreciate the authors' rebuttal. Unfortunately, I don't find their arguments persuasive.
> >
> > "Joint optimization is therefore not introduced merely because it is possible, but because posterior inference is the underlying task for which Bayesian adaptive design is optimized."
> >
> > in BED, we seek a policy that produces highly informative actions w.r.t. the belief state. It is not mathematically necessary to explicitly construct this belief state from the information set at test time in order to yield such actions; I agree that it is plausible that designing a training procedure to also learn a representation of the true posterior along with the EIG maximizing actions would lead to improved performance, but it is not obvious to me why that would be the case in the event that this is true.
> >
> > If we somehow do require that our model can quickly provide accurate posterior representations in addition to the optimal actions, then I would appreciate some references that make it clear when that would really be the case.
> >
> > I am open to being persuaded on the latter; the former I feel less malleable on but am curious to hear the authors' perspective once more.
> >
> >
> > I do appreciate the inclusion of Wasserstein distances.
> >
> > For now I keep my score.

---

> > > ### Author Response · Authors · 2026-04-06
> > >
> > > We thank the reviewer again for engaging with the core motivation of the paper. It seems that our central claim may still have come across as stronger than intended.
> > >
> > > We agree that, in one important line of BED work, the goal is to learn a policy that produces highly informative designs without necessarily constructing the posterior explicitly [1, 2]. We therefore do not claim that posterior inference, or even posterior estimation, is necessary in order to obtain informative designs.
> > >
> > > Our claim is narrower and more practical. In adaptive likelihood-free BED, posterior inference is often itself part of the downstream deployment objective, and in many such settings it is also a **computational bottleneck**. Examples include sensor placement, model calibration, MRI acquisition, and Bayesian inverse problems [3-6]. Thus, jointly amortizing design and inference can be useful even when it is not strictly required in principle. Our goal is not to introduce posterior inference “because it can be done,” but to learn a policy that leads to the most informative posterior object the scientist ultimately wants to query after each step.
> > >
> > > This is also the practical limitation of methods that focus only on design optimization. They can avoid posterior estimation during training, but they still leave the posterior inference problem to be solved at deployment time. This perspective is also reflected in work that explicitly connects SBI and BED: [7] study BED for models with intractable likelihoods in a setting where inference is the ultimate goal; [8] explicitly position their work as bridging SBI and BED; [9] motivate their method through the computational difficulty of handling adaptive design and posterior inference together; and [10] is another very recent example of the field moving in this direction.
> > >
> > > Empirically, we do provide evidence for this practical value: Experiment 3 tests whether a learned adaptive policy improves over a non-adaptive/random designs in a high-dimensional setting, and it does (see Table 2, e.g. $0.177 \pm 0.001$ vs. $2.056 \pm 0.007$). In that sense, we do not see joint training as a combinatorial gimmick. Instead, **it improves amortized posterior inference with arbitrary generative models in settings with sequential measurements**. This sets the stage for really interesting but also challenging applications, such as adaptive optics of proposing light shaping patterns [11] or wavefront shaping to increase signal-to-noise ratio of in-vivo imaging [12], both with lifetime estimation as their downstream posterior inference task.
> > >
> > > If this narrower interpretation of our contribution is unambiguous, we would kindly ask the reviewer to reconsider whether the paper’s significance may be closer to that recognized by the rest of the reviewers.
> > >
> > > ---
> > >
> > > [1] Foster et al. (2021), “Deep Adaptive Design: Amortizing Sequential Bayesian Experimental Design”
> > >
> > > [2] Ivanova et al. (2021), “Implicit Deep Adaptive Design: Policy-Based Experimental Design without Likelihoods”
> > >
> > > [3] Beck et al. (2018), “Fast Bayesian experimental design: Laplace-based importance sampling for the expected information gain”
> > >
> > > [4] Baptista et al. (2024), “Bayesian model calibration for block copolymer self-assembly: Likelihood-free inference and expected information gain computation via measure transport”
> > >
> > > [5] Orozco et al. (2024), “Probabilistic Bayesian optimal experimental design using conditional normalizing flows”
> > >
> > > [6] Aretz et al. (2024), “A greedy sensor selection algorithm for hyperparameterized linear Bayesian inverse problems with correlated noise models”
> > >
> > > [7] Drovandi and Pettitt (2013), “Bayesian Experimental Design for Models with Intractable Likelihoods”
> > >
> > > [8] Zaballa and Hui (2025), “Optimizing Likelihoods via Mutual Information: Bridging Simulation-Based Inference and Bayesian Optimal Experimental Design”
> > >
> > > [9] Iollo et al. (2025), “Bayesian Experimental Design via Contrastive Diffusions”
> > >
> > > [10] Klein et al. (2026), “Supercharging Simulation-Based Inference for Bayesian Optimal Experimental Design”
> > >
> > > [11] Nizam et al. (2025), “A Novel Technique for Fluorescence Lifetime Tomography”
> > >
> > > [12] Yu et al. (2022), “Wavefront shaping: A versatile tool to conquer multiple scattering in multidisciplinary fields”

---

### Official Review · Reviewer_WQVK · 2026-03-13

**Soundness:** 3
**Presentation:** 3
**Significance:** 3
**Originality:** 3
**Overall Recommendation:** 4
**Confidence:** 3

**Summary:**

This paper proposes a unified framework, JADAI, to jointly amortize Bayesian adaptive design and posterior inference. It trains three separate networks end-to-end using a generic posterior loss that acts as a proxy for total expected information gain (TEIG): for normalized density models, the TEIG can be written as a telescoping sum of per-step log-posterior difference, while JADAI generalizes it by replacing log-posterior with generic posterior loss. This frees the framework from requiring tractable likelihoods, enabling the use of models without exact log-density computation (e.g. diffusion models) for high-dimensional and complex posteriors. JADAI shows superior or competitive performance across several adaptive design benchmarks.

**Compliance With Llm Reviewing Policy:**

Affirmed.

**Final Justification:**

The rebuttal addressed my main concerns, and I believe the planned additions can strengthen the method and the clarity of presentation. I will remain my original positive assessment.

**Key Questions For Authors:**

1. Theoretical justification of the generic loss \- see weakness section.
2. Would be good to see whether a shared backbone for policy and summary networks benefits the performance.
3. Could you show some examples of scalling JADAI to even higher dimensional problems?

**Limitations:**

Yes. The authors discussed several limitations including: 1) expensive posterior sampling, 2) dependence on differentiable simulators, 3) separate networks for the policy, summary, and posterior.

**Strengths And Weaknesses:**

**Strength**

1. **Flexibility in posterior estimator**. The generic posterior loss free JADAI from architectural constraints, thus it can leverage either normalizing flow or diffusion models for posterior estimation.
2. **Strong empirical results**. JADAI shows competitive or better performance across three benchmarks, including one high-dimensional image discovery task.
3. **Detailed training engineering choices**. The paper presents sensible engineering tricks including truncated nested backpropagation, curriculum rollout length and design-prior mix-in, with ablation results to validate these designs.

**Weakness**

1. **The generic loss lacks a clear theoretical connection to the TEIG lower bound**. For diffusion models, the proxy loss is no longer the same as the BA variational lower bound. The paper claims the objective "still encourages trajectories along which the posterior loss decreases”, while a formal theoretical connection would strengthen the claim.
2. **Requiring differentiable simulators**. JADAI's end-to-end gradient flow requires differentiable simulators, which limits its applicability compared to RL-based alternatives, which can handle black-box settings.
3. **Baseline comparison is not incomplete**. The comparison is limited to a few shared experimental setups.

---

> ### Author Rebuttal · Authors · 2026-03-31
>
> # JADAI and BA bound
> We thank the reviewer for raising this important question about the relationship between JADAI’s training objective and the TEIG lower bound. Please find the sketch below, and we will provide a more detailed description in a new appendix section.
> JADAI's detached telescoping (eq. 16-18) produces the $\psi$-gradient $\sum_{t=0}^T \nabla_\psi \mathbb{E}[\ell_t]$, decomposing the training signal into per-horizon posterior losses. Each term relates to the TEIG at horizon $t$ via the BA bound [1, 2]:
>
> $$\mathrm{TEIG}_t = \mathrm{sCEE}_t(\psi) + \delta_t(\psi) \quad \text{with} \quad \delta_t(\psi) := \mathbb{E} \left[\mathrm{KL}\left(p(\theta \mid h_t) \Vert q\_\psi (\theta \mid h_t) \right)\right] \geq 0.$$
>
> which holds for any posterior approximation regardless of training method.
>
> In the density case ($\ell_t = -\log q_\psi$), each $\nabla_\psi \mathbb{E}[\ell_t] = \nabla_\psi \delta_t(\psi)$, so JADAI exactly minimizes the summed BA gaps across all horizons.
> As noted in Section 3.2, this log-density ratio interpretation no longer holds directly in the score-based case ($\ell_t = \mathcal{L}{\mathrm{post}}^{\mathrm{diff/flow}}$). However, the detached telescoping still accumulates $\nabla_\psi \ell_t$ across all $t \leq T$, serving as a surrogate that reduces each $\delta_t(\psi)$ and tightens $\mathrm{sCEE}_t(\psi)$ toward $\mathrm{TEIG}_t$. Besides the regular component in $\delta_t$ that stems from using a neural network for approximating the density, the score-to-density component in $\delta_t$ can be reduced over training [3, 4, 5], and $\mathrm{sCEE}_t(\psi) \leq \mathrm{TEIG}_t$ remains valid regardless.
>
> [1] Foster et al. (2021), “Deep Adaptive Design: Amortizing Sequential Bayesian Experimental Design”
>
> [2] Blau et al. (2023), “Statistically Efficient Bayesian Sequential Experiment Design via Reinforcement Learning with Cross-Entropy Estimators”
>
> [3] Song et al. (2021), “Score-Based Generative Modeling through Stochastic Differential Equations”
>
> [4] Song et al. (2021), “Maximum Likelihood Training of Score-Based Diffusion Models”
>
> [5] Lu et al. (2022), “Maximum Likelihood Training for Score-Based Diffusion ODEs
> by High-Order Denoising Score Matching”
>
> ---
>
> # Shared policy and history
> We agree that tighter coupling between the policy and history modules is a promising future direction (see e.g., ALINE [1]). In JADAI, however, this is not a drop-in modification, since the policy selects $\xi_t$ from $h_{t-1}$ before the new observation is available, while the history network produces $h_t$ only after incorporating $(\xi_t, x_t)$. A shared-backbone variant would need to be designed around these different roles during rollout.
>
> [1] Huang et al. (2025), “ALINE: Joint Amortization for Bayesian Inference and Active Data Acquisition”
>
> ---
>
> # Scaling to higher dimensions
> We share the reviewer’s interest in understanding how far JADAI can be pushed to even higher-dimensional settings. The current submission already includes image discovery precisely to evaluate the framework beyond low-dimensional (in the context of SBI) targets, and the reviewer also recognized this experiment as a high-dimensional benchmark. More extensive scaling studies, especially for larger target or design spaces, would be valuable but require a dedicated setup rather than a small addition. We see this as important follow-up work.
>
> ---
>
> # Differentiable simulators
> We agree with the reviewer that the need for simulator differentiability is a limitation of JADAI’s current applicability, as we also pointed out in the conclusion section. However, we want to distinguish between limitations and weaknesses of the manuscript. In our view, this assumption limits the class of simulators the current method can be applied to, but it is not a flaw in the technical contribution itself.
>
> We also do not see this as a severe limitation of the setting, since differentiable simulation and surrogate modeling are quite popular in practice. For example, libraries such as TorchOptics [1] and Chromatix [2] already offer fully differentiable optics simulations, and packages such as AutoEmulate [3] can build general differentiable surrogates. More broadly, recent literature [4] emphasizes the growing importance of differentiable programming. Prompted by your comment, we will update the discussion to better reflect the current limitation and the realistic path toward relaxing it in the future.
>
> [1] Filipovich et al. (2024), “TorchOptics: An open-source Python library for differentiable Fourier optics simulations”
>
> [2] Deb et al. (2025), “Chromatix: a differentiable, GPU-accelerated wave-optics library”
>
> [3] Stoffel et al. (2025), “AutoEmulate: A Python package for semi-automated emulation”
>
> [4] Lavin et al. (2021), “Simulation Intelligence: Towards a New Generation of Scientific Methods”
>
> # Baseline comparison Table 1
> We thank the reviewer for pointing out the missing values in the baseline comparison and refer to the answer for MLg2.

---

> > ### Author Rebuttal · Reviewer_WQVK · 2026-04-04
> >
> > Thank you for the rebuttal. I believe the proposed additions - the formal connection between JADAI's training objective and the BA bound, the clarifications on baseline comparisons, and the expanded discussion of limitations from differentiable simulators - will strengthen the manuscript. My major concerns have been addressed, and I am happy to maintain my positive score.

---

> > > ### Author Response · Authors · 2026-04-06
> > >
> > > We thank the reviewer again for their thoughtful engagement during the rebuttal phase. Their comments helped us improve both the clarity and the overall strength of the manuscript.

---

### Decision · Program_Chairs · 2026-04-30

**Decision:**

Accept (regular)

**Comment:**

This work proposes a framework for jointly amortizing adaptive experimental design and posterior inference in likelihood-free settings. The framework is based on a generic posterior-loss objective that enables flexible inference models within adaptive design, which is technically sound. The method shows promising empirical performance across several test problems.

Reviewer WQVK and Reviewer MLg2 are positive and support acceptance, noting strong empirical results and a solid contribution, with concerns largely addressed in the rebuttal. Reviewer wH8J is also supportive after rebuttal, raising mainly clarity and exposition issues. Reviewer Hf83 is negative, primarily questioning the practical motivation and significance of jointly optimizing design and inference, and remains unconvinced after rebuttal.

The main concerns across reviewers relate to clarity and positioning. In particular, the theoretical connection to the TEIG objective is less clear outside the density-based case, although this was largely clarified in the rebuttal. In addition, the practical motivation for joint optimization is somewhat underspecified in the main text. While the rebuttal provides a reasonable justification (settings where posterior inference is required at deployment), this is not as clearly demonstrated empirically as it could be.

Overall, I find this to be a solid contribution with promising ideas, but with limitations in theoretical clarity and practical justification that reduce its impact. I therefore recommend a weak accept.